# Conformation selection by ATP-competitive inhibitors and allosteric communication in ERK2

Jake W Anderson[1†], David Vaisar[1], David N Jones[2], Laurel M Pegram[1‡], Guy P Vigers[3§], Huifen Chen[4], John G Moffat[4], Natalie G Ahn[1*]

[1]Department of Biochemistry, University of Colorado, Boulder, United States; [2]Department of Pharmacology, University of Colorado Anschutz Medical Center, Boulder, United States; [3]ArrayBioPharma, Inc., Boulder, United States; [4]Genentech, Inc., South San Francisco, United States

*For correspondence:
natalie.ahn@colorado.edu

Present address: [†]Department of Biochemistry, Oxford University, Oxford, United Kingdom; [‡]Loxo Oncology, Louisville, United States; [§]Allium Consulting LLC, Boulder, United States

**Abstract** Activation of the extracellular signal-regulated kinase-2 (ERK2) by phosphorylation has been shown to involve changes in protein dynamics, as determined by hydrogen-deuterium exchange mass spectrometry (HDX-MS) and NMR relaxation dispersion measurements. These can be described by a global exchange between two conformational states of the active kinase, named 'L' and 'R,' where R is associated with a catalytically productive ATP-binding mode. An ATP-competitive ERK1/2 inhibitor, Vertex-11e, has properties of conformation selection for the R-state, revealing movements of the activation loop that are allosterically coupled to the kinase active site. However, the features of inhibitors important for R-state selection are unknown. Here, we survey a panel of ATP-competitive ERK inhibitors using HDX-MS and NMR and identify 14 new molecules with properties of R-state selection. They reveal effects propagated to distal regions in the *P*+1 and helix αF segments surrounding the activation loop, as well as helix αL16. Crystal structures of inhibitor complexes with ERK2 reveal systematic shifts in the Gly loop and helix αC, mediated by a Tyr-Tyr ring stacking interaction and the conserved Lys-Glu salt bridge. The findings suggest a model for the R-state involving small movements in the N-lobe that promote compactness within the kinase active site and alter mobility surrounding the activation loop. Such properties of conformation selection might be exploited to modulate the protein docking interface used by ERK substrates and effectors.

## eLife assessment

This **fundamental** study provides **compelling** evidence to explain how chemical variations within a set of kinase inhibitors drive the selection of specific Erk2 conformations. Conformational selection plays a critical role in targeting medically relevant kinases such as Erk2 and the findings reported here open new avenues for designing small molecule inhibitors that block the active site while also steering the population of the enzyme into active or inactive conformations. Since protein dynamics and conformational ensembles are essential for enzyme function, this work will be of broad interest to those working in drug development, signal transduction, and enzymology.

## Introduction

The extracellular signal-regulated protein kinases, ERK1/2, are key components of the MAP kinase signaling pathway controlling cell proliferation, survival, morphology, and migration (*Lavoie et al., 2020*; *Roskoski, 2012*; *Plotnikov et al., 2011*). Due to the prevalence of oncogenic mutations in Ras, RAF, and MEK, components of this pathway have been targeted for the development of successful

anti-cancer therapies (*Halle and Johnson, 2021*; *Tangella et al., 2021*; *Canon et al., 2019*; *Ryan et al., 2020*). Although inhibitors targeting ERK1/2 have not advanced to clinical use, clinical trials are ongoing for several, including ulixertinib/BVD523 (Biomed Valley Diagnostics), ravoxertinib/GDC0994 (Genentech), and tizakertib/ATG017/AZD0364 (Antengene), for the treatment of cancers with activating mutations in KRAS, BRAF, NRAS, MEK1/2, and ERK1/2 (*Grierson et al., 2023*; *Goodwin et al., 2023*; *Wu et al., 2021*; *Varga et al., 2020*; *Kong et al., 2023*; *Chen et al., 2021*). Importantly, reversible ATP-competitive inhibitors targeting ERK1/2 have been shown to be effective in cancer cells and tumors with acquired resistance to BRAF and MEK inhibitors (*Ryan et al., 2015*; *Morris et al., 2013*; *Hatzivassiliou et al., 2012*; *Roskoski, 2019*; *Pan et al., 2022*). Therefore, future anticancer drug development would benefit from a more complete understanding of the properties of ERK inhibitors.

Many protein kinases undergo relatively large conformational changes when they switch from inactive to active states. Such changes lead to the repositioning of conserved active site residues, including a Lys-Glu salt bridge in the N-terminal lobe between strand β3 and helix αC that hydrogen bonds with nucleotide phosphate, an Asp residue in the DFG motif that coordinates $Mg^{2+}$, and an Asp in the HRD motif that catalyzes base-assisted phosphoryl transfer to the substrate hydroxyl acceptor (*Taylor et al., 2019*; *Amatya et al., 2019*). By contrast, ERK2 is unusual in that only small conformational changes occur within the active site between the inactive, unphosphorylated (0P) and the active, dual phosphorylated (2P) forms of the kinase (*Figure 1*). X-ray structures show that dual phosphorylation of ERK2 at T183 and Y185 remodels the activation loop by forming phosphate salt bridges with several Arg residues (*Zhang et al., 1994*; *Canagarajah et al., 1997*). This accommodates the S/T-P recognition motif common to ERK substrates and exposes a pocket for binding a hydrophobic docking motif found in substrates, scaffolds, and other effectors (*Lee et al., 2004*). But within the ATP binding site, the conserved residues that form the K52-E69 salt bridge, the DFG metal binding site (D165), and the HRD catalytic base (D147) are largely overlapping (*Figure 1C*). Thus, the changes in the active site that accompany ERK2 activation are small and difficult to discern crystallographically.

Solution measurements revealed that the activation of ERK2 is accompanied by changes in protein dynamics. NMR Carr-Purcell-Meiboom-Gill (CPMG) relaxation dispersion measurements showed that dual phosphorylation results in a global exchange behavior that can be modeled by two conformational states, named 'R' and 'L,' which exchange on a millisecond timescale (*Xiao et al., 2014*; *Xiao et al., 2015*; *Iverson et al., 2020*). Thus, 2P-ERK2 can be characterized by the interconversion of at least two conformers. Whereas the L-state resembles that seen with 0P-ERK2, the R-state appears only in 2P-ERK2, where it exchanges with an R:L population ratio of 80:20 at 25 °C. Crystal structures of nucleotide-bound kinase show significant differences between 0P- and 2P-ERK2. In 2P-ERK2, the ATP analog, AMP-PNP, is positioned close to the catalytic base and in-line for nucleophilic attack on the Pγ phosphate (*Pegram et al., 2019*; *Lechtenberg et al., 2017*). By contrast, ATP bound to 0P-ERK2 is markedly distorted, moving Pγ away from the base by more than 6 Å (*Zhang et al., 2012*; *Smorodinsky-Atias et al., 2016*). Thus, despite the structural similarity between the apoenzyme forms, 2P-ERK2 promotes ATP binding interactions that are more productive for catalytic function. The unique appearance of the R-state in 2P-ERK2 has been proposed to confer the ability to form a Michaelis complex poised for phosphoryl transfer, while L may represent a state that promotes ADP product release (*Pegram et al., 2019*).

Interestingly, ATP competitive inhibitors of ERK show properties of conformation selection, where ligand binding traps conformational states in 2P-ERK2 that normally undergo reversible exchange in the apoenzyme. Binding of the inhibitor, Vertex-11e (VTX11e), to 2P-ERK2 shifts the L↔R equilibrium completely to the R-state, while the inhibitor SCH772984 shifts the equilibrium completely to the L-state (*Rudolph et al., 2015*; *Pegram et al., 2019*). Conformation selection is absent in the inhibitor, GDC0994, where R and L retain a population ratio similar to that seen in the apoenzyme (*Pegram et al., 2019*). This selective recognition could explain why the binding of VTX11e is stronger to 2P-ERK2 than 0P-ERK2 (*Rudolph et al., 2015*). In contrast, GDC0994 and SCH772984 bind both active and inactive kinase forms with equal affinity (*Pegram et al., 2019*; *Ward et al., 2019*).

HDX-MS also reveals differences between VTX11e, GDC0994, and SCH772984 in their ability to induce perturbations at the *P*+1 segment (*Pegram et al., 2019*). *P*+1 is distal to the kinase active site, and together with the activation loop forms part of the interface for substrate and effector binding. This suggests that R↔L conformational shifts within the active site can propagate out to the activation loop and surrounding regions. NMR-CPMG measurements confirm this, by showing that residues in

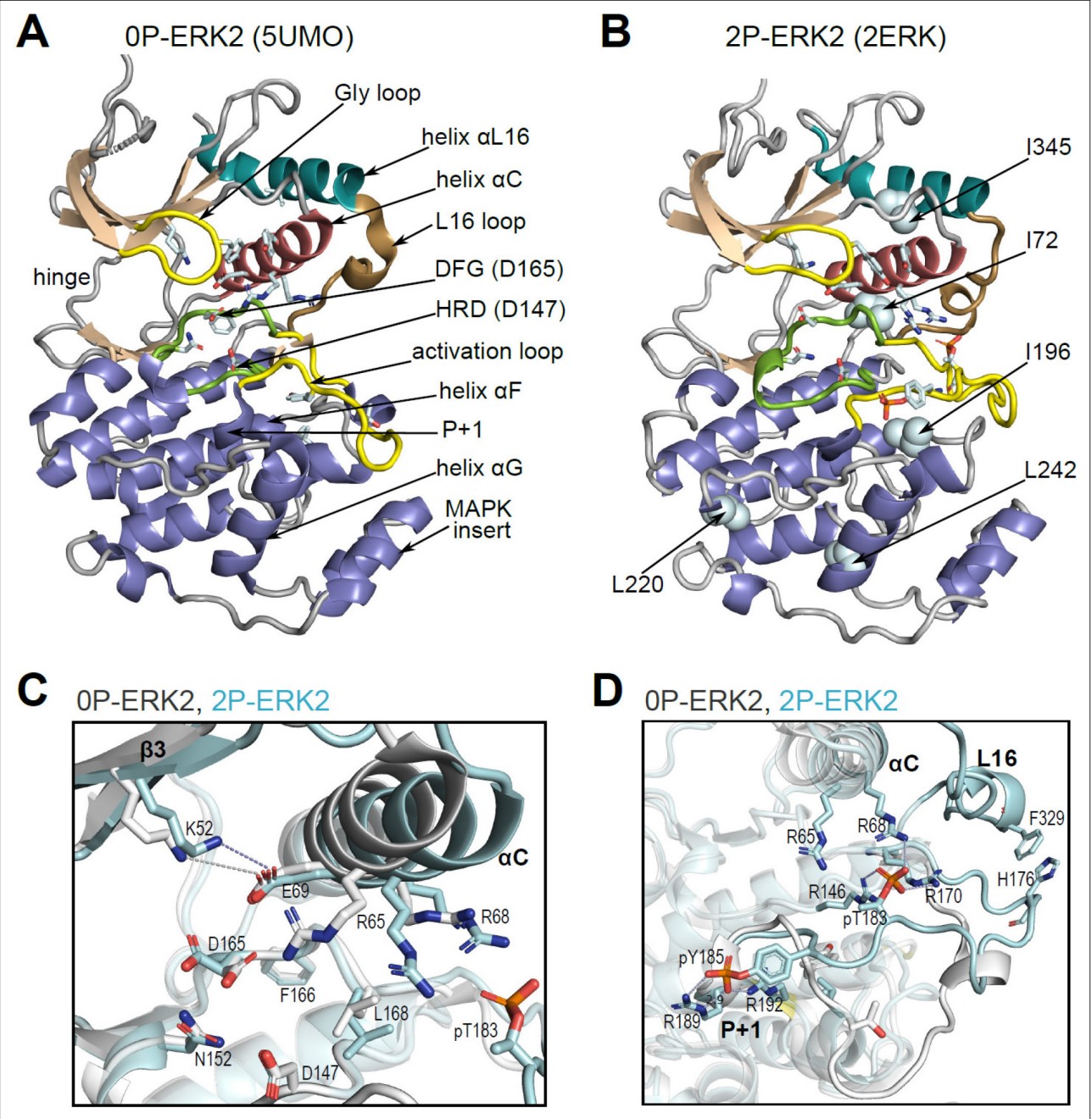

**Figure 1.** Structural features of extracellular signal-regulated kinase-2 (ERK2). (**A, B**) X-ray structures of (**A**) 0P-ERK2 (PDBID:5UMO) and (**B**) 2P-ERK2 (PDBID:2ERK) apoenzymes. Panel **A** labels conserved motifs in ERK2 common to protein kinases. Panel **B** labels residues that illustrate L↔R exchange (I72, L220, L242; see *Figure 2* and *Figure 7*), and key chemical shift perturbations (I196, I345; see *Figure 8*). (**C**) Structural superposition of 0P-ERK2 (white) and 2P-ERK2 (pale blue), illustrating overlapping positions of active site residues and an outward shift of helix αC upon dual phosphorylation. Root-mean-square deviations between 0P- and 2P-ERK2 for catalytic site residues were K52-0.69 Å, E69-0.16 Å, D147-0.055 Å, D165-0.88 Å. (**D**) Superposition of 0P-ERK2 and 2P-ERK2, illustrating conformational differences in the activation loop. In 2P-ERK2, pT183, and pY185 form ion pairs with multiple Arg residues, while the L16 loop folds into a 3/10 helix with side chain interactions to the activation loop. Structures were superpositioned by aligning Cα atoms within the C-terminal domain (residues 109–141, 205–245, 272–310).

the activation loop participate in the same global exchange motion as residues surrounding the active site (*Iverson et al., 2020*). Therefore, solution measurements of conformation selective inhibitors reveal allosteric coupling between the active site and distal regions around the activation loop.

A consequence of this behavior is that conformation-selective inhibitors can alter the interactions of ERK2 with other proteins. For example, the binding of VTX11e partially inhibits the rate of dephosphorylation at the activation loop by the MAP kinase phosphatase, MKP3/DUSP6, while SCH772984 enhances this rate (*Pegram et al., 2019*). The features of VTX11e controlling its property of R-state selection are unknown. An attempt to address this compared crystal structures of complexes formed between 2P-ERK2 and VTX11e (100% R), GDC0994 (80% R), or SCHCPD336, a close analog of SCH772984 (0% R) (*Pegram et al., 2019*; *Chaikuad et al., 2014*; *Blake et al., 2016*). But the results were inconclusive. Binding of SCHCPD336 disrupted Gly loop and helix αC interactions leading to a disordered activation loop, suggesting that the L-state might communicate N-lobe conformational changes to the activation loop dynamics. In contrast, VTX11e and GDC0994 formed similar close contacts with active site residues with no major distortions in the ATP pocket. Given the limited set of compounds with small structural differences, it was difficult to identify features of binding that could explain why VTX11e is selective for the R-state but GDC0994 is not.

Here, we survey a panel of small molecules with varying chemistries, using HDX-MS and NMR to classify their properties of conformation selection. Of the 19 inhibitors examined, 14 behave in a similar manner to VTX11e, with conformational selection for the R-state. These include 13 novel inhibitors and ulixertinib/BVD523, a promising ERK inhibitor with blood-brain barrier permeability (*Sullivan et al., 2018*; *Yu et al., 2022*; *Sigaud et al., 2023*). Four inhibitors resemble GDC0994 in allowing R↔L exchange with populations comparable to apoenzyme, and one allows exchange but partially shifts the equilibrium toward the R-state. NMR chemical shifts reveal perturbations unique to R-state inhibitors located in the *P*+1 segment and helix αL16, and remote from the inner-sphere region of ligand contact. Co-crystal structures of R-state selective inhibitors complexed with ERK2 show small but systematic conformational shifts in the Gly loop, helix αC, and helix αL16, in which the Gly loop moves closer to the DFG segment, while helicesαC and αL16 shift outwards. This movement rotates the N- and C-lobes into a more closed configuration and moves the K52-E69 salt bridge closer to the conserved DFG motif. Such small but systematic conformational shifts suggest a working model for R-state selection, involving movements of the N-lobe that favor a more compact active site with allosteric coupling to the activation loop.

## Results

### Solution measurements reveal R-state conformation selection

NMR experiments were used to examine conformation selection by inhibitor binding. Previously, NMR-CPMG measurements of ERK2, selectively [methyl-$^{13}$C,$^1$H]-labeled on Ile, Leu, and Val (ILV) residues, showed that phosphorylation introduced global motions involving multiple residues in the N- and C-lobes surrounding the active site, as well as in the activation loop (*Xiao et al., 2014*; *Iverson et al., 2020*). These could be modeled by a global two-state exchange process (R↔L) with an exchange rate constant $k_{ex}$ = 300 s$^{-1}$. As shown previously (*Xiao et al., 2014*; *Iverson et al., 2020*), the R:L populations were 80:20 at 25 °C and 50:50 at 5 °C for the 2P-ERK2 apoenzyme, and 0:100 for the 0P-ERK2 apoenzyme at both 25°C and 5°C.

NMR spectroscopy was used to test the effect of inhibitors on conformational exchange in 2P-ERK2. 2D ($^{13}$C,$^1$H)-HMQC spectra were collected on complexes of ERK2 with BVD523 and VTX11e, which share chemical features including a central amido-linked pyrrole scaffold, and GDC0994, which has a distinct central pyridone scaffold (*Figure 2A*; *Figure 2—figure supplement 1*; *Figure 2—figure supplement 2*). Each inhibitor was added to ERK2 in saturating amounts (1:1.2 ERK2:inhibitor), as illustrated for VTX11e and GDC0994 (*Figure 2—figure supplement 3*; *Figure 2—figure supplement 4*). *Figure 2B–D* illustrates methyl probes with slow exchange (Δω>>$k_{ex}$), where the relative populations of R and L separated by their chemical shift differences can be observed in HMQC NMR spectra. Saturation binding of BVD523 shifted the R↔L equilibrium to 100% R at both 25°C and 5°C, matching the effects of VTX11e (*Figure 2B–D*). In contrast, GDC0994 had little effect on the R and L populations relative to apoenzyme (*Figure 2B–D*). The results show that BVD523, like VTX11e, exhibits properties

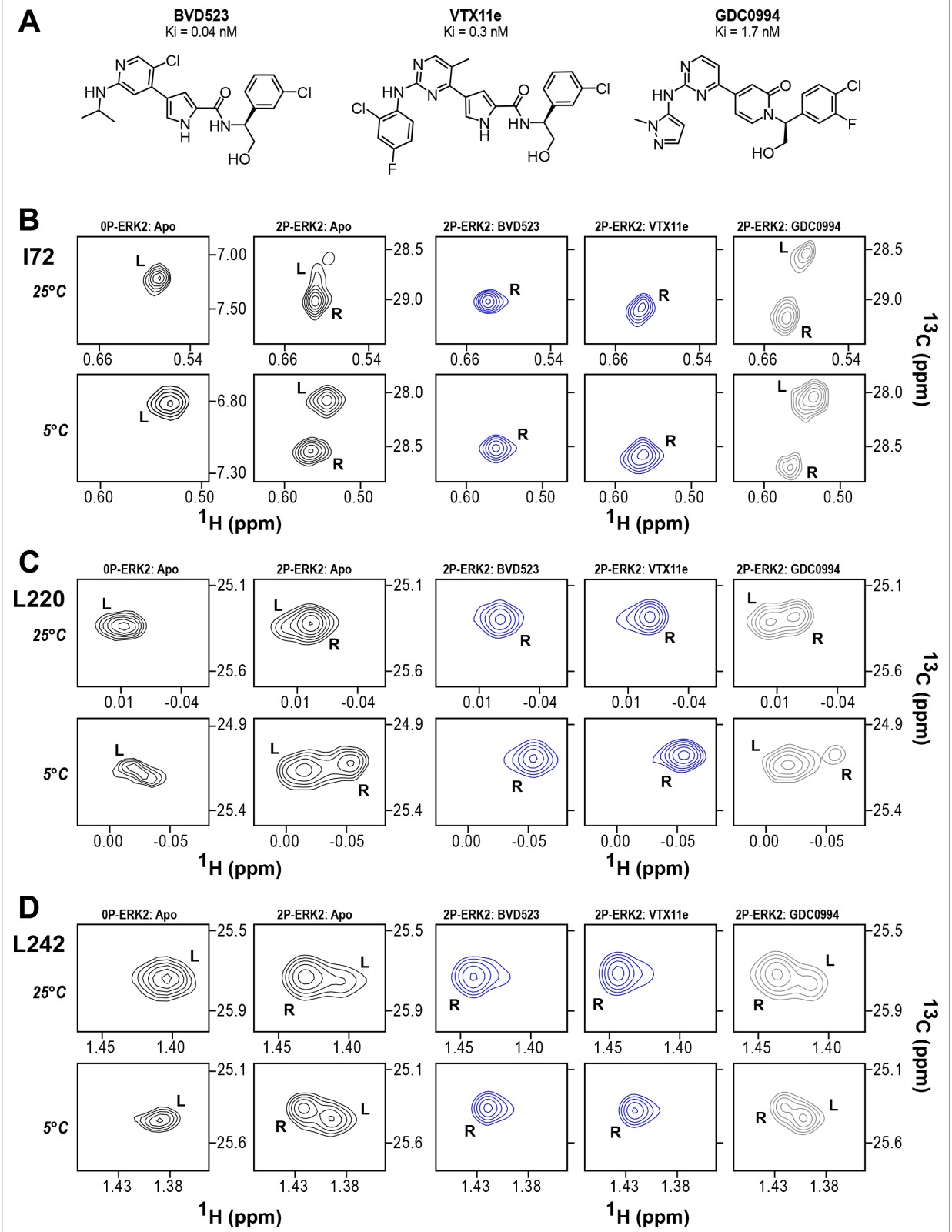

**Figure 2.** BVD523 and Vertex-11e (VTX11e) stabilize the R-state in 2P-ERK2. (**A**) Chemical structures of the ATP-competitive ERK1/2 inhibitors, BVD523, VTX11e and GDC0994. (**B–D**) 2D-HMQC spectra collected at 25°C and 5°C ([ERK2]:[inhibitor]=1.0:1.2), showing [methyl-$^{13}$C,$^{1}$H] peaks of residues (**B**) I72, (**C**) L220 and (**D**) L242, which report R and L conformers. Their locations in the ERK2 structure are shown in *Figure 1*. 2P-ERK2 complexed with BVD523 and VTX11e (shown in blue) shift to 100% R at all temperatures, while 2P-ERK2 complexed with GDC0994 (shown in gray) retains conformational

*Figure 2 continued on next page*

*Figure 2 continued*

exchange between R:L populations of 80:20, similar to the apoenzyme. Previous studies show that 0P-ERK2 complexed with all inhibitors retains the L conformer seen in the apoenzyme (*Pegram et al., 2019*). Full NMR spectra are shown in *Figure 2—figure supplement 1* and *Figure 2—figure supplement 2*. Titration of 2P-ERK2 with VTX11e and GDC0994 to demonstrate binding saturation is shown in *Figure 2—figure supplement 3* and *Figure 2—figure supplement 4*.

The online version of this article includes the following figure supplement(s) for figure 2:

**Figure supplement 1.** 2D-HMQC NMR spectra of apoenzyme and inhibitor-bound extracellular signal-regulated kinase-2 (ERK2).

**Figure supplement 2.** 2D-HMQC NMR spectra of apoenzyme and inhibitor-bound extracellular signal-regulated kinase-2 (ERK2).

**Figure supplement 3.** Titration of Vertex-11e (VTX11e) binding to 2P-ERK2.

**Figure supplement 4.** Titration of GDC0994 binding to 2P-ERK2.

of R-state conformational selection. This behavior might explain why BVD523 preferentially binds the activated kinase, with ~10 fold higher affinity for 2P-ERK2 than 0P-ERK2 (*Germann et al., 2017*).

HDX-MS experiments were conducted as a complementary approach to compare the effects of BVD523 to VTX11e and GDC0994 on conformational selection (*Supplementary file 1*). Binding of all three inhibitors reduced deuteration at conserved regions in the active site, including the Gly-rich loop, helix αC, the hinge, helix αE, and helix αL16 to a comparable degree between 0P- and 2P-ERK2 (*Figure 3A–C*, light green; *Figure 3—figure supplement 1*; *Figure 3—figure supplement 2*; *Figure 3—figure supplement 3A–E and G*). All three inhibitors also reduced HDX in the β7-β8 loop, although only in 2P-ERK2 (*Figure 3A–C*, dark green; *Figure 3—figure supplement 1*; *Figure 3—figure supplement 3F*). Such patterns reflect steric protection from solvent due to the occupancy of ligands in the nucleotide-binding site. In contrast, the effects of inhibitors differed in the DFG motif and adjacent β9 strand, where binding of BVD523 and VTX11e led to greater HDX protection compared to GDC0994 (*Figure 3A–C*, blue; *Figure 3D*; *Figure 3—figure supplement 4*). BVD523 and VTX11e also induced HDX protection greater than GDC0994 in regions farther from the active site, including the *P*+1 segment and helix αF (*Figure 3A–C*, blue; *Figure 3E and F*). These effects at *P*+1 and helix αF reveal allosteric coupling between the active site and distal regions surrounding the activation loop. Thus, solution measurements reported conformation selection for the R-state by BVD523 and VTX11e, but not GDC0994. The similarities in chemical structure between BVD523 and VTX11e suggested that features in their central scaffold and/or right-side regions might contribute to their shared properties of R-state selection.

## Conformation selection among small molecule inhibitors of ERK1/2

To expand the structural diversity of studied compounds, we surveyed a panel of 17 small molecule inhibitors of ERK2 (*Figure 4A*) that had been produced in the course of chemical development of GDC0994 (*Blake et al., 2016*). All of these were ATP-competitive, and all inhibited ERK1/2 with subnanomolar $K_i$. The compounds were chosen for variations in their left-side region, which interacts with the kinase hinge and solvent channel; their central scaffold, which contacts hydrophobic residues from strands β7-β8 and the Gly loop; and their right-side region, which interacts with the Gly loop. The left-side varied between pyrazole and tetrahydropyran groups, the central scaffold alternated pyridone and triazolopyrazine ring structures, and the right-side accommodated many groups with different sizes and chemistries (*Figure 4B*). Residue contacts with each region are illustrated for VTX11e and GDC0994 in *Figure 4C*.

HDX-MS was used to examine the effect of each inhibitor on the conformational mobility of ERK2. Thirteen inhibitors (#2, #3, #5-12, #14, #16, #17) produced HDX patterns resembling BVD523 and VTX11e, with strong protection of the DFG motif in both 0P- and 2P-ERK2, and protection of the P+1 segment and helix αF in 2P-ERK2 (*Figure 5A-C*; *Figure 5—figure supplement 1*). In contrast, three inhibitors (#4, #13, #15) showed HDX patterns resembling GDC0994, indicating lesser protection of the DFG motif and no protection in the regions surrounding the activation loop (*Figure 5A-C*; *Figure 5—figure supplement 2*). One inhibitor (#1) showed strong protection of DFG, but weak protection of the activation loop regions compared to BVD523/VTX11e (*Figure 5A-C*). The effect of each inhibitor on deuterium uptake was quantified for these peptide segments by summing the differences in centroid mass between apo- and ligand-bound enzymes over all time points. This yielded a difference area measurement (dAUC), which was then ranked for each segment. *Figure 6*

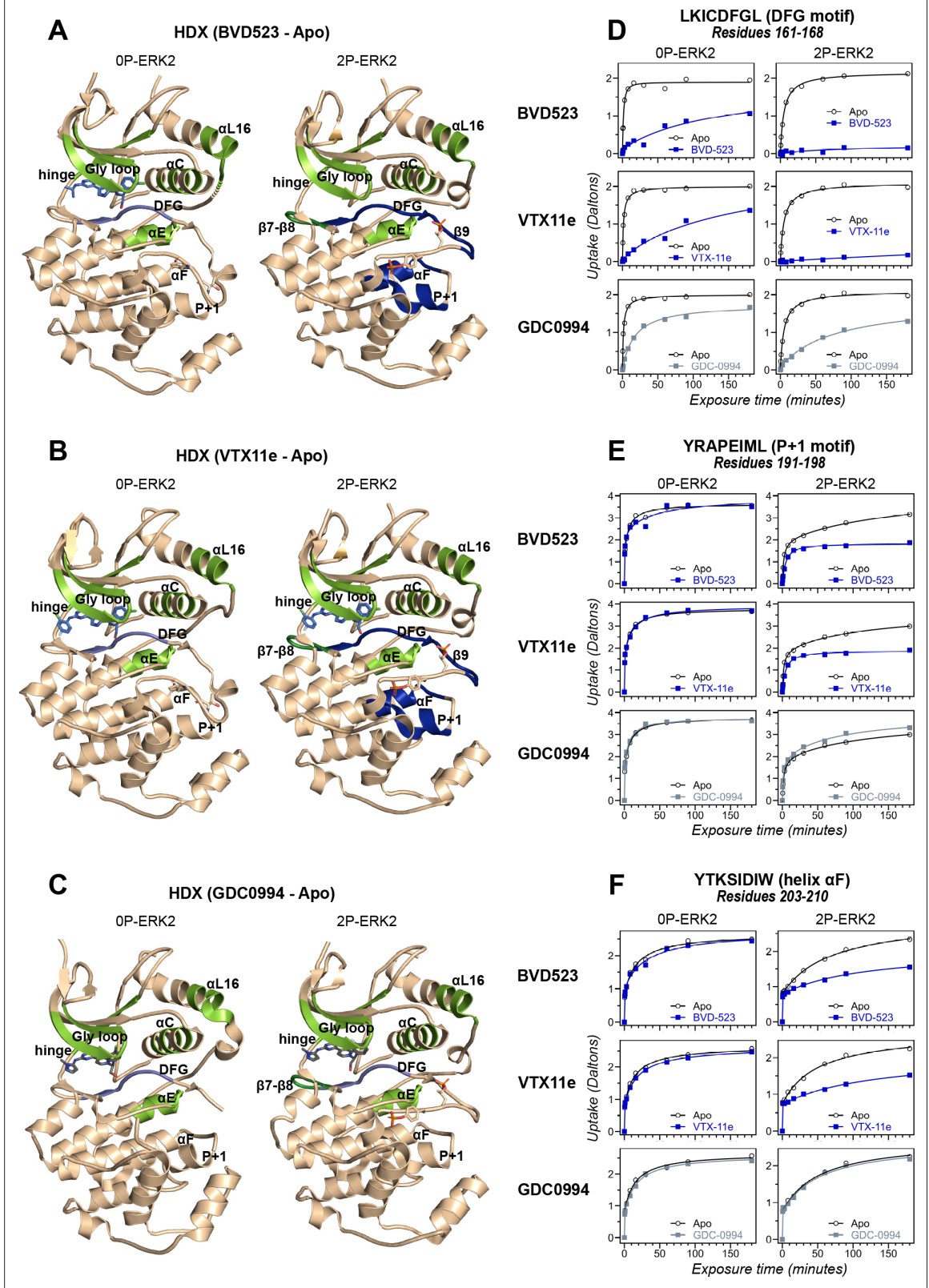

**Figure 3.** Binding of R-state inhibitors reveals allosteric coupling between the active site and the activation loop. (**A–C**) Summary of hydrogen-deuterium exchange (HDX) experiments indicating regions that change in deuterium uptake upon binding of (**A**) BVD523, (**B**) Vertex-11e (VTX11e), and (**C**) GDC0994. (**D–F**) HDX time courses showing effects of inhibitors on deuterium uptake at the (**D**) DFG motif (peptide 161–168: LKICDFGL), (**E**) P+1 segment (peptide 191–198, YRAPEIML), and (**F**) helix αF (peptide 203–210: YTKSIDIW). Colored segments in panels **A–C** indicate regions where

*Figure 3 continued on next page*

*Figure 3 continued*

HDX decreases or increases upon binding each inhibitor at saturating concentration ([ERK2]:[inhibitor]=1.0:1.2). Full peptide coverage and locations of segments that undergo changes in HDX with inhibitor binding are shown in *Figure 3—figure supplement 1* and *Figure 3—figure supplement 2*. Highlighted in light green in panels **A–C** are regions where a similar degree of HDX protection is seen with all inhibitors in both 0P-ERK2 and 2P-ERK2 (Gly loop, hinge, helices αC, αE, and αL16). HDX protection is similar with all inhibitors in strands β7-β8 (dark green) in 2P-ERK2, but not 0P-ERK2. Time courses for these peptides are shown in *Figure 3—figure supplement 3*. Highlighted in light blue are regions where BVD523, VTX11e, or GDC0994 lead to decreased HDX uptake around the DFG motif, in 0P-ERK2 or 2P-ERK2. Highlighted in dark blue are regions where BVD523 and VTX11e lead to increased HDX protection, compared to GDC0994. These occur only in 2P-ERK2, and include the DFG motif and adjacent strand β9, as well as the *P*+1 segment and helix αF. Time courses for strand β9 are shown in *Figure 3—figure supplement 4*. Full HDX datasets for all inhibitors are presented in *Figure 1*. Crystal structures shown in panels **A–C** are (**A**) PDBID: 6GDQ (left) and 2ERK (right); (**B**) PDBID: 4QTE (left) and 6OPK (right); (**C**) PDBID: 5K4I (left); and 6OPH (right).

The online version of this article includes the following figure supplement(s) for figure 3:

**Figure supplement 1.** Proteolytic peptides analyzed by hydrogen-deuterium exchange mass spectrometry (HDX-MS).

**Figure supplement 2.** Structural map of regions showing hydrogen-deuterium exchange (HDX) responses to inhibitor binding.

**Figure supplement 3.** Hydrogen-deuterium exchange (HDX) time courses of regions with comparable responses to different inhibitors.

**Figure supplement 4.** Hydrogen-deuterium exchange (HDX) time courses with differential responses to inhibitor binding.

shows rankings plotted to compare the DFG and P+1 peptides. The results showed good agreement between the behavior of each segment, with obvious separation in both dimensions between thirteen inhibitors clustering with BVD523 and VTX11e, three clustering with GDC0994, and one in between.

The HDX-MS results indicated that many compounds that are structurally distinct from BVD523 and VTX11e nevertheless share properties of R-state selection when bound to 2P-ERK2, while others resemble GDC0994 in maintaining conformational exchange. This was confirmed for a subset of seven inhibitors by HMQC NMR, chosen to represent variations in left-side, central scaffold, and right-side substituents. Methyl probes reporting R↔L exchange in 2P-ERK2 showed that four inhibitors that clustered with BVD523 and VTX11e by HDX also shifted populations to 100% R by NMR, while two inhibitors that clustered with GDC0994 maintained R:L population ratios of approximately 80:20 (*Figure 7A–C*; *Figure 7—figure supplement 1*; *Figure 7—figure supplement 2*). Inhibitor #1, which appeared between the two main clusters in *Figure 6*, shifted to the R-state, but only to a partial degree that still allowed exchange (*Figure 7A–C*). Thus, the HDX and NMR solution measurements are in good agreement and revealed R-state selection in 2P-ERK2 among a majority of the 17 compounds surveyed, even though they are chemically distinct from VTX11e and BVD523.

## NMR chemical shifts report inhibitor chemistry and long-distance allostery

Next, we examined NMR chemical shifts to identify [methyl-$^{13}$C,$^{1}$H]-ILV probes responsive to binding the different inhibitors (*Figure 8—figure supplement 1*; *Supplementary file 2*), as distinct from the methyl probes above reporting R↔L exchange. Chemical shift perturbations from apoenzyme were examined for correlations with different inhibitor features (*Figure 8*). All inhibitor complexes led to peak disappearances attributed to broadening (indicated by red asterisks below the x-axes in *Figure 8—figure supplement 1*). These occurred at residues expected to be in direct contact with ligands, based on kinase-inhibitor structures (*Roskoski, 2016*). They mapped to strand β7 (L154) below the inhibitor binding site, and the αC-β4 loop (I82) which contacts the inhibitor from the back pocket (*Figure 8*, black; see *Figure 4C*). One residue forming Gly loop contacts (V37), as well as second-shell residues in strand β5 (I101) and the N-terminal end (V12) appeared broadened with all inhibitors except VTX11e and BVD523 (*Figure 8*).

Four methyl probes showed chemical shifts that could be correlated with the identify of the left-side substituent, which in ERK2:GDC0994 and ERK2:VTX11e are located near the hinge and interact with the solvent-exposed front pocket (*Figure 4C*; *Pegram et al., 2019*; *Chaikuad et al., 2014*; *Blake et al., 2016*). *Figure 8* shows the locations of these residue probes in green. Here, chemical shift perturbations separated the inhibitors #4, #6, #8, and #15, which share tetrahydropyran at the left-side position, from #1, #5, #16, and GDC0994, which share pyrazole. Three methyl probes (L105, L155, L161) were located near residues that directly contact the inhibitor left-side, and one (L26) was located in strand β1, nearby a hydrophobic side chain (I29) that contacts the pyrazole ring in

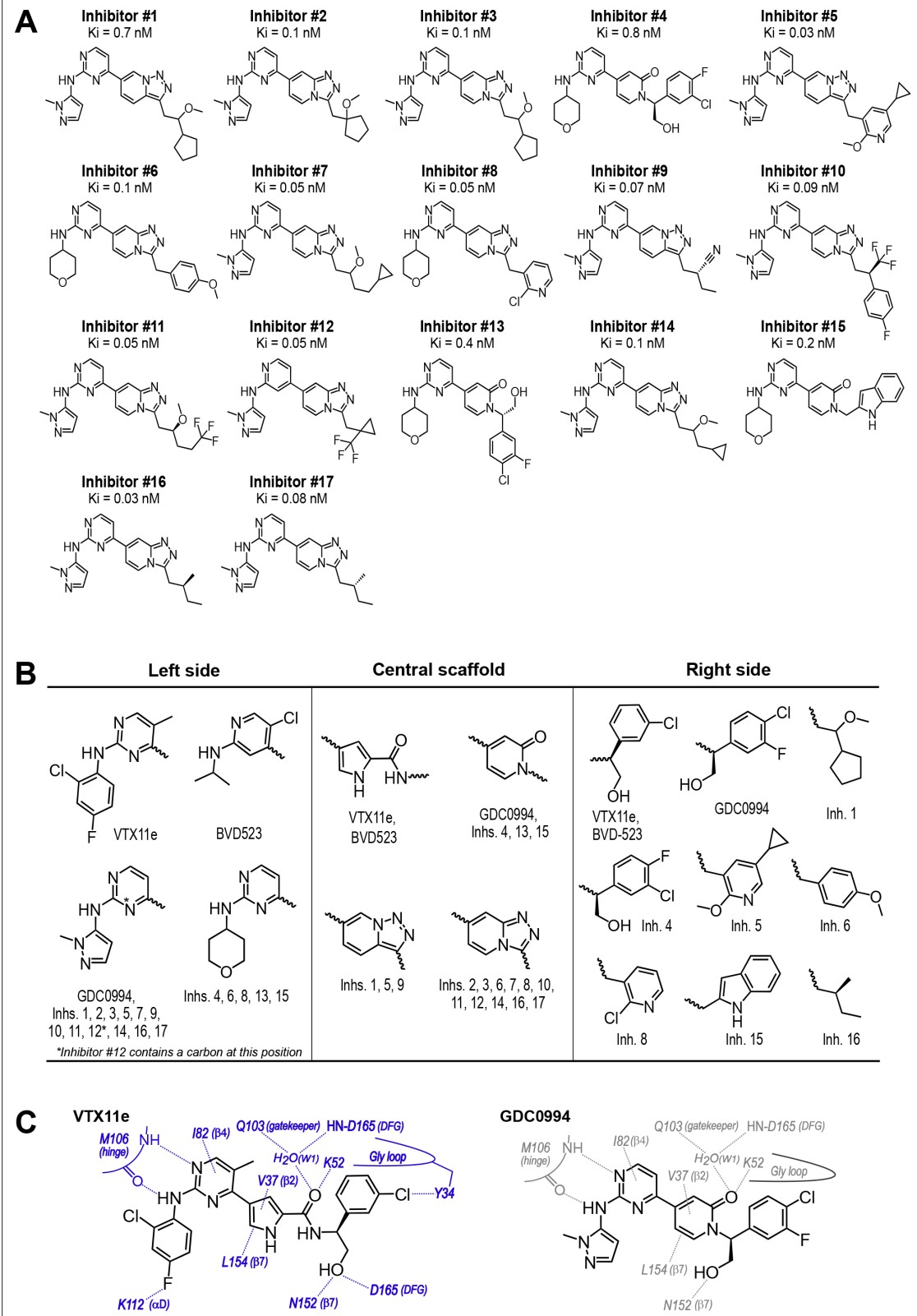

**Figure 4.** Novel ATP-competitive inhibitors of ERK1/2. (**A**) A panel of 17 novel ERK1/2 inhibitors was surveyed in this study. Estimates of $K_i$ were measured using kinase assays for phosphorylation of Omnia peptide substrate (Invitrogen). (**B**) Variations among ERK1/2 inhibitors in their left-side, central scaffold, and a sampling of right-side substituents. (**C**) Contacts formed by VTX11e and GDC0994 with active site residues in 2P-ERK2, based on published X-ray structures (PDBID:6OPK, PDBID:6OPH).

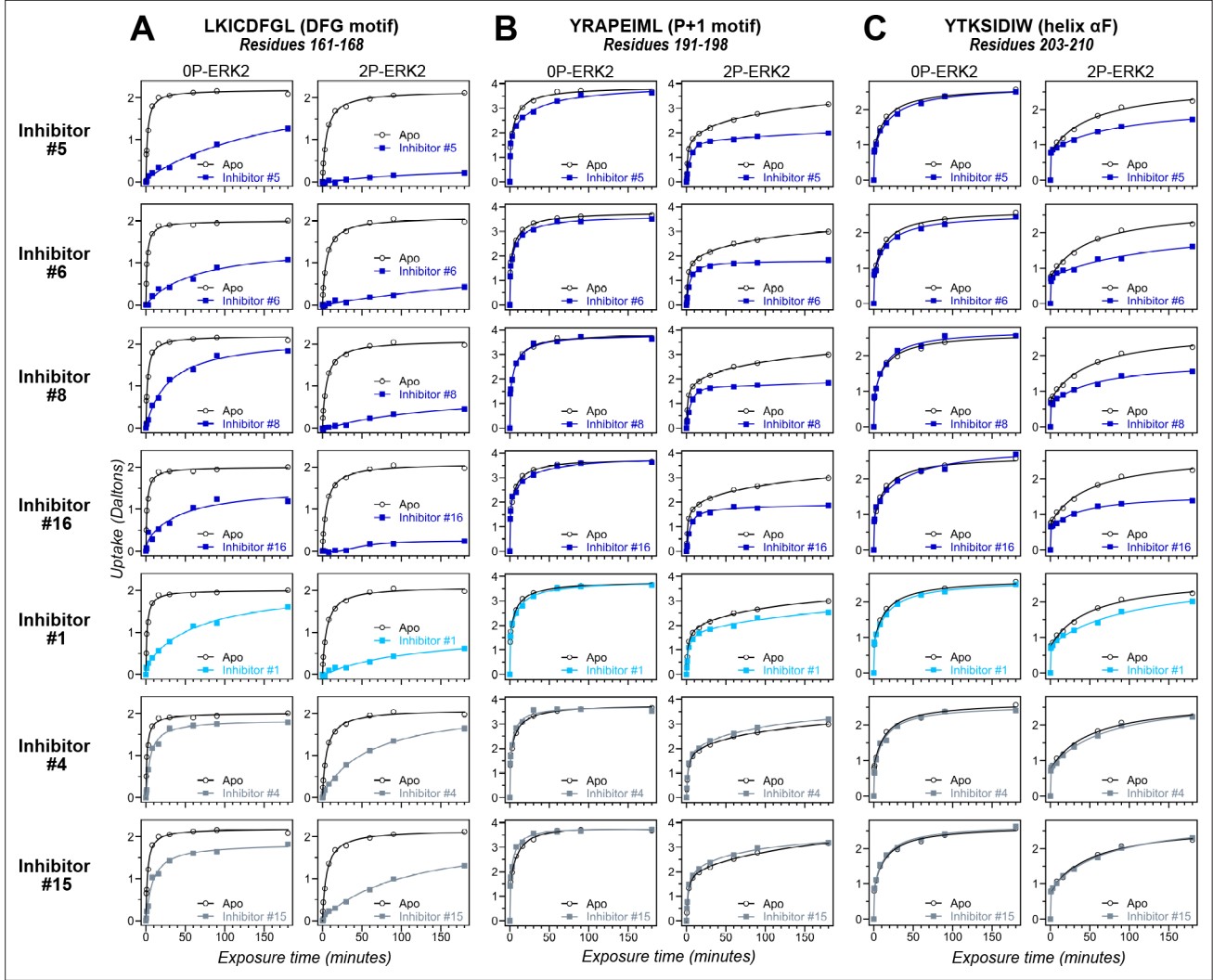

**Figure 5.** Hydrogen-deuterium exchange (HDX) assays survey conformation selection among compounds in the extracellular signal-regulated kinase-2 (ERK) inhibitor panel. HDX measurements were performed with representative inhibitors shown in *Figure 4*, chosen for variations in their left-side, central scaffold, and right-side substituents. Time courses show deuterium uptake at the (**A**) DFG motif (peptide 161–168: LKICDFGL), (**B**) *P*+1 segment (peptide 191–198, YRAPEIML), and helix αF (peptide 203–210: YTKSIDIW). Enhanced HDX protection (strongly decreased uptake) in each segment by inhibitors #5, #6, #8, and #16 (blue) suggest properties of conformation selection for the R-state, while lower protection by inhibitors #4 and #15 (gray) suggest retention of conformational exchange. Inhibitor #1 (cyan) shows HDX properties intermediate to these two groups. HDX time courses for the full set of 17 inhibitors are shown in *Figure 5—figure supplement 1* and *Figure 5—figure supplement 2*.

The online version of this article includes the following figure supplement(s) for figure 5:

**Figure supplement 1.** Differential hydrogen-deuterium exchange (HDX) responses to new inhibitors were surveyed in this study.

**Figure supplement 2.** Differential hydrogen-deuterium exchange (HDX) responses to new inhibitors were surveyed in this study.

ERK2:GDC0994 (*Pegram et al., 2019*; *Blake et al., 2016*). Thus, several peaks displayed chemical shifts or broadening consistent with direct binding interactions.

In contrast, two probes showed chemical shift behaviors that correlated with R-state conformation selection (*Figure 8*, red). These residues were located away from the active site, within the *P*+1 segment (I196) and the C-terminal helix αL16 (I345). Here, R-state selective inhibitors (VTX11e, BVD523, #5, #6, #8, #16) induced greater chemical shift perturbations, while inhibitors that retained conformational exchange (GDC0994, #4, #15) were closer to apoenzyme. The location of I196 was consistent with the HDX patterns showing greater protection in the *P*+1 loop by each of the R-state selective inhibitors. The location of I345 in helix αL16 suggested an association of the R-state with N-lobe perturbations proximal to helix αC, which interacts with αL16 (*Figure 1*). Thus, NMR identified changes occurring

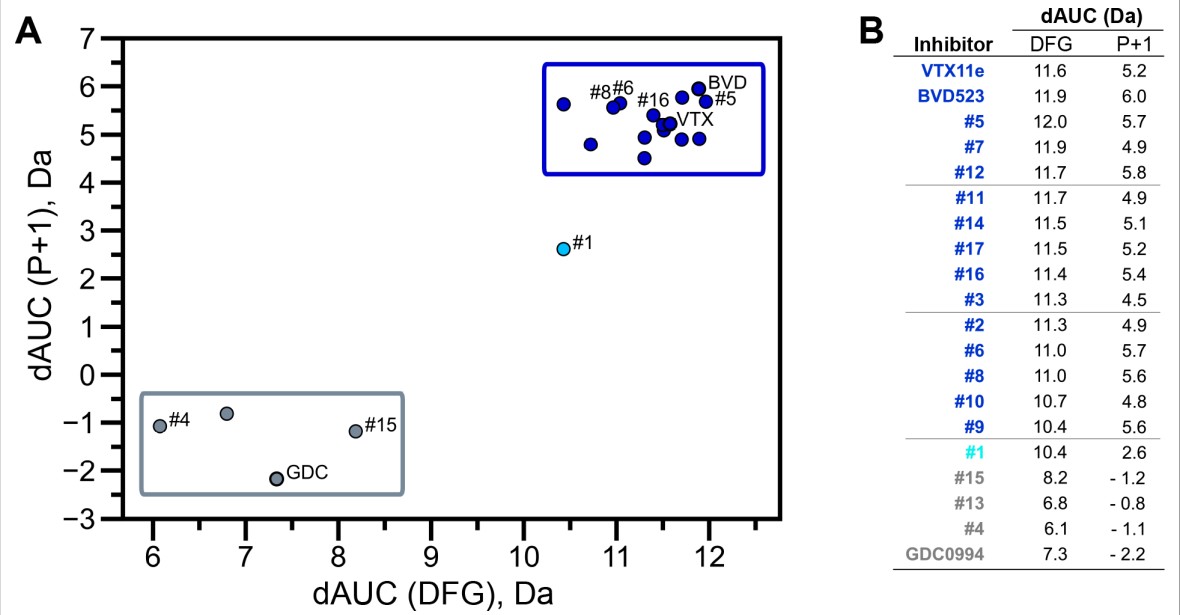

**Figure 6.** Extracellular signal-regulated kinase (ERK) inhibitors differentially cluster with Vertex-11e (VTX11e)/BVD523 or GDC0994. Effects of inhibitors on hydrogen-deuterium exchange (HDX) were quantified by a difference AUC measurement, which calculates dAUC = $\Sigma_t(HDX_{Apo}- HDX_{Inhibitor})_t$ over all time points. (**A**) Plot comparing dAUC for the DFG motif (peptide 161–168) *vs* P+1 segment (peptide 191–198) reveals 13 inhibitors clustered with VTX11e/BVD523, 3 inhibitors clustered with GDC0994, and one inhibitor with intermediate properties. Labels mark the representative set of inhibitors shown in *Figure 4*. (**B**) Difference area measurement (dAUC) values for each inhibitor.

remotely from the ligand binding site that were sensitive to shifts towards the R-state, as reported by HDX-MS. The results helped confirm that the binding of the new R-state inhibitors to the ATP pocket couples to changes in distal regions, including those around the activation loop.

## R-state selection properties correlate with conformational shifts in the N-lobe

We asked if structural features accompanying R-state selection might be observable by X-ray crystallography. Structures of 2P-ERK2 complexed with inhibitors #8 and #16 were solved (*Supplementary file 3*) and compared to previous structures of 2P-ERK2 complexed with VTX11e and GDC0994 (*Pegram et al., 2019*). Both inhibitors #8 and #16 formed contacts with active site residues that matched those contacting VTX11e and GDC0994 (*Figure 9A*). They included hydrogen bonds between the left-side aminopyrimidine and the hinge backbone (M106) and tetrahydropyran with the front pocket (K112); direct or water-mediated hydrogen bonds between the central triazolopyridine scaffold and the gate-keeper (Q103), Lys-Glu salt bridge (K52), and DFG motif (D165 backbone amide); and van der Waals interactions between the right-side chlorophenyl or isopentane groups and the Gly loop.

While conformational differences between each of the inhibitor complexes were small, systematic shifts nevertheless appeared in the N-lobe, involving the Gly loop, helix αC, and helix αL16. In the co-crystal structure with VTX11e, these elements shifted outwards (away from the ligand and towards the activation loop) relative to GDC0994 (*Figure 9B*). This difference could be attributed to a residue contact unique to VTX11e, whose right-side 3-chlorobenzyl substituent forms a Cl-π contact with Y34 on the Gly loop (*Figure 9B* left panel, see *Figure 4C*). Y34 in turn forms π-π ring-stacking interactions with Y62 on helix αC, suggesting cooperative ternary anion-π-π interactions (*Lucas et al., 2016*). These contacts, together with the K52-E69 salt bridge, likely couple movements of the β1-β2-β3 sheet to helix αC, which in turn promotes movement of the adjoining helix αL16. This results in the repositioning of helix αC and αL16 to accommodate the bulky chlorobenzyl group on VTX11e and moves the Gly loop to a more closed conformation. In contrast, the corresponding 3-fluoro 4-chlorobenzyl substituent on GDC0994 is positioned higher in the binding site and tilted upwards into the Gly loop relative to VTX11e (*Figure 9B*, right panel). This holds the Gly loop in a more open conformation and shifts the position of helix αC and αL16 inwards (towards the ligand). Thus, the variations between

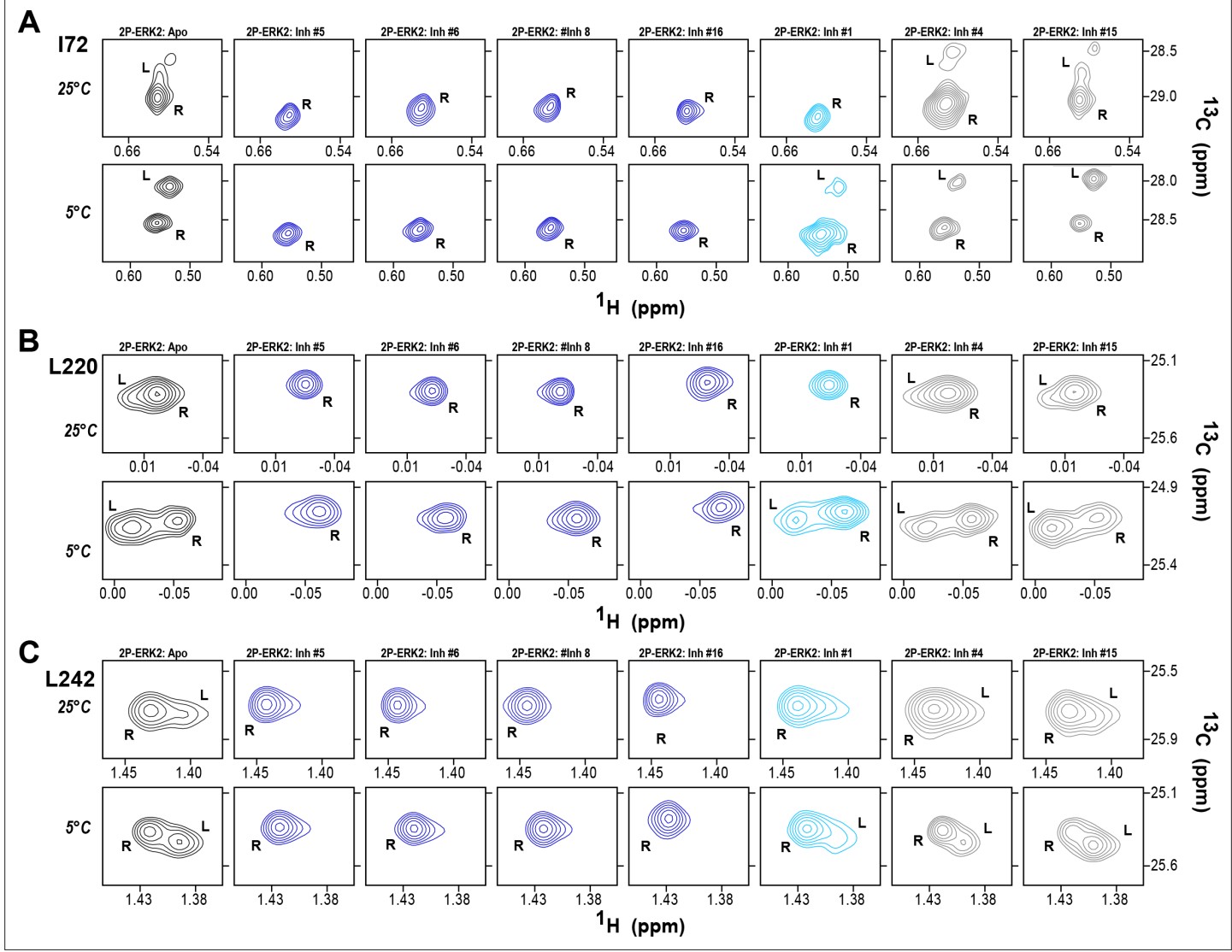

**Figure 7.** 2D-HMQC NMR confirms R-state selection by extracellular signal-regulated kinase (ERK) inhibitors. 2D-HMQC NMR spectra were collected on 2P-ERK2 at 25°C and 5°C. Effects on [methyl $^{13}$C,$^1$H] peaks of residues (**A**) I72, (**B**) L220, and (**C**) L242 are shown for the representative set of inhibitors in *Figure 5*. Inhibitors #5, #6, #8, and #16 (shown in blue) shifted the R↔L equilibrium to 100% R at all temperatures, confirming R-state selection as suggested by their HDX behaviors in *Figure 5* and *Figure 6*. Inhibitors #4 and #15 (in gray) retained R:L populations comparable to those with GDC0994. Inhibitor #1 (in cyan) showed partial selection for the R state but retained conformational exchange, as evidenced by an L-state population present at 5 °C. Full NMR spectra are shown in *Figure 7—figure supplement 1* and *Figure 7—figure supplement 2*.

The online version of this article includes the following figure supplement(s) for figure 7:

**Figure supplement 1.** 2D-HMQC NMR spectra of inhibitor-bound 2P-ERK2.

**Figure supplement 2.** 2D-HMQC NMR spectra of inhibitor-bound 2P-ERK2.

right-side substituents and their interactions with the Y34-Y62 stack underlie small conformational shifts in the N-lobe that affect domain opening and closure.

Like VTX11e, complexes of 2P-ERK2 with inhibitors #8 and #16 (PDBID: 8U8K, 8U8J) showed outward shifts of the Y34-Y62 stack, the Gly loop, and helices αC and αL16, relative to GDC0994 (*Figure 9C and D*, left panel). Unlike VTX11e, the 2-chloropyridine and isopentane substituents on inhibitors #8 and #16 respectively were located more than 6 Å from Y34. Both contacted the backbone of strand β1 at angles that allowed but were unlikely to force Gly loop closure (*Figure 9C and D*, right panels). Inspection of their triazolopyridine central scaffold showed interactions that were distinct from the pyridone scaffold in GDC0994. In GDC0994, the pyridone carbonyl oxygen formed

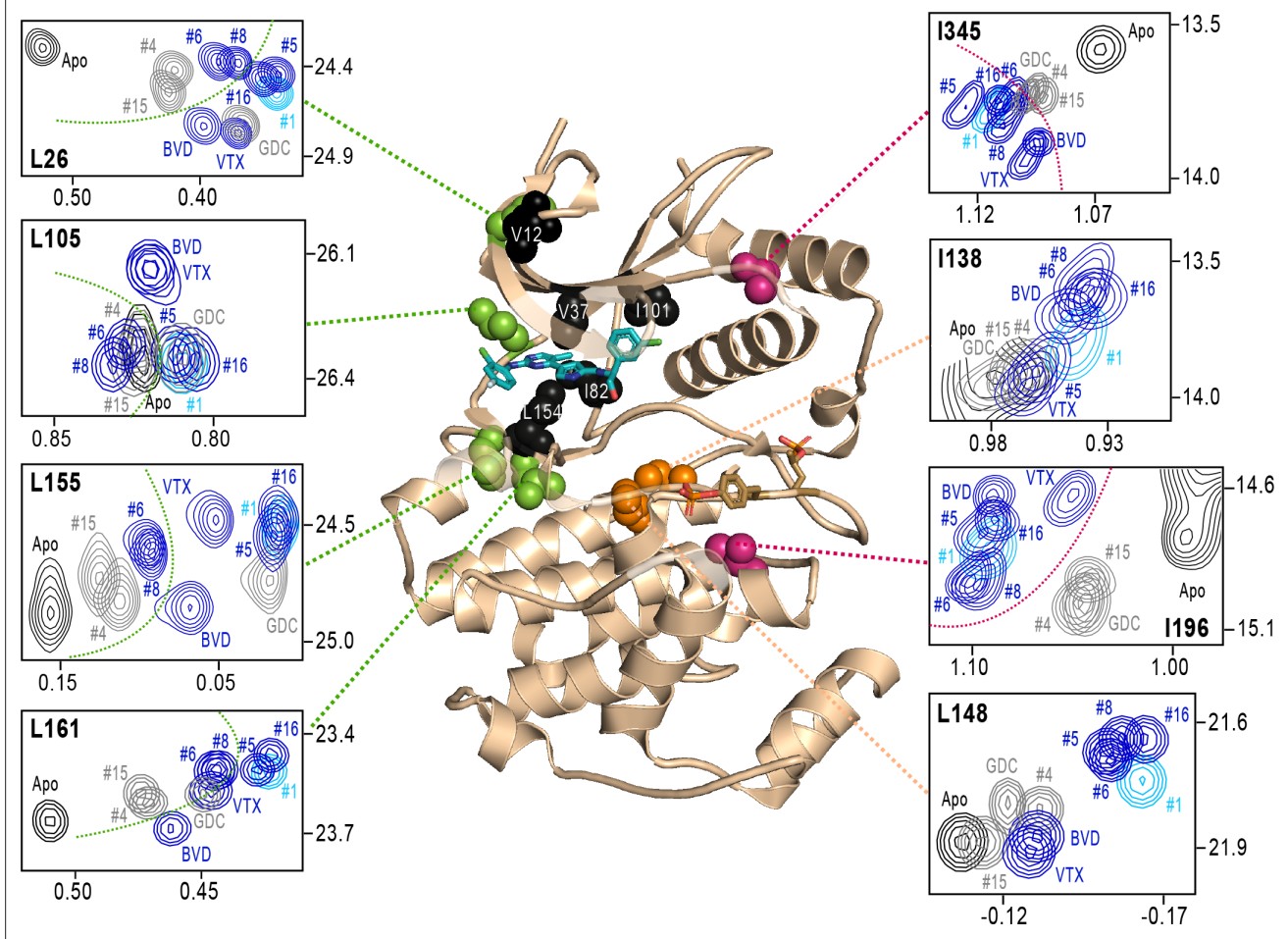

**Figure 8.** NMR chemical shift patterns reveal long-distance perturbations by R-state inhibitors. 2D-HMQC NMR spectra of 2P-ERK2 at 25 °C showing [methyl $^{13}$C,$^1$H] peaks with significant changes in chemical shift upon binding extracellular signal-regulated kinase (ERK) inhibitors. NMR peaks colored blue correspond to inhibitors characterized as R-state selective (Vertex-11e : VTX11e, BVD523, #5, #6, #8, #16). Peaks colored gray correspond to inhibitors that allow conformational exchange (GDC0994, #4, #15). Peaks in cyan correspond to inhibitor #1, which shows partial R-state selection. Shown in red spheres on the X-ray structure (PDBID:6OPK) are residues (I196, I345) that separate the R-state selective inhibitors from those that allow exchange. Green spheres are residues (L26, L105, L155, L161) that separate inhibitors with left-side tetrahydropyran (#4, #6, #8, #15) from other substituents. Black spheres are residues that broaden upon binding all inhibitors (I82, L154), or all inhibitors except VTX11e (V12, V37, I101); These together with residues in orange (I138, L148) reflect first- and second-sphere regions of contact around the binding site. Chemical shifts and chemical shift perturbations (Δδ) for all assigned residues are shown in *Figure 8—figure supplement 1* and *Supplementary file 2*.

The online version of this article includes the following figure supplement(s) for figure 8:

**Figure supplement 1.** Chemical shift changes induced by inhibitor binding to 2P-ERK2.

hydrogen bonds with both K52 and a bound water (W1) coordinated to the gatekeeper side chain (Q103) and the DFG main chain amide (D165) (*Figure 9E*). In contrast, the two nitrogens in the triazole ring of inhibitors #8 and #16 formed separate hydrogen bonds to K52 and W1, respectively. This separation had the effect of shifting the K52-E69 salt bridge position relative to GDC0994. As a result, the Gly loop shifts to a more closed state, allowing the pocket below the Gly loop to accommodate two water molecules that form OH-π interactions with Y34, and favoring outward movement of helicesαC and αL16 (*Figure 9C and D*). In this way, the R-state inhibitors VTX11e, #8, and #16 share small but systematic movements that lead to Gly loop closure and movements of helixαC and αL16 in the direction of the activation loop. Differences were also seen in the hydrogen bond network around the conserved DFG region. Here, a second bound water (W2) formed a bridge between the catalytic residues K52 and D165 in complexes with VTX11e, #8, and #10, but was absent in GDC0994 (*Figure 9B and D*). This revealed a relayed network of hydrogen bond interactions linking the DFG motif to the

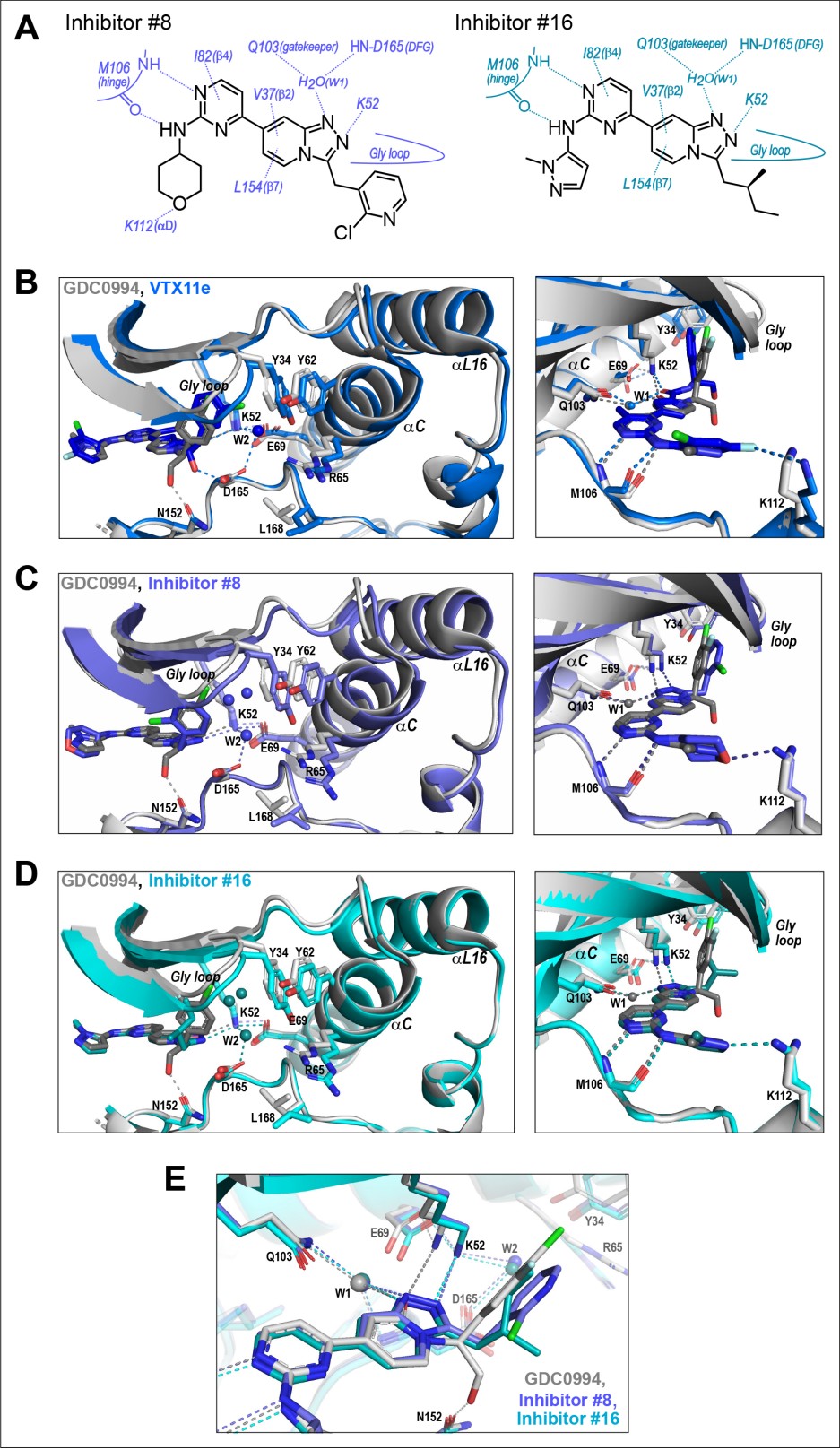

**Figure 9.** R-state inhibitors promote outward movements of N-lobe structural elements. (**A**) Summary of contacts formed by inhibitors #8 and #16 with active site residues in co-crystal structures with 2P-ERK2 (PDBID: 8U8K and 8U8J). (**B**) The active site of 2P-ERK2 complexed with GDC0994 (PDBID:6OPH, gray) and Vertex-11e (VTX11e) (PDBID:6OPK, blue) (*Pegram et al., 2019*). *Left panel:* Front view showing the movement of the Gly loop, helix

*Figure 9 continued on next page*

*Figure 9 continued*

αC, and helix αL16 in an 'outward' direction (away from the inhibitor) upon binding of VTX11e (blue), relative to GDC0994 (gray). The movement can be attributed to the right-side 3-chlorobenzyl substituent in VTX11e which interacts with theπ orbital of Y34 in the Gly loop (Cl-π distance, 3.5 Å). In turn, π–π stacking interactions between Y34 and Y62 couples movements of the Gly loop to helix αC. *Right panel:* Side view showing left-side hydrogen bond contacts with main chain atoms of hinge residue M106, as typical of ATP-competitive kinase inhibitors. (**C, D**) Active site of 2P-ERK2 complexed with GDC0994 (PDBID:6OPH, gray) and (**C**) inhibitor #8 (PDBID:8U8K, slate), or (**D**) inhibitor #16 (PDBID:8U8J, cyan). Like VTX11e, inhibitors #8 and #16 move the Gly loop, helix αC, and helix αL16 outward, relative to GDC0994. Right panels (**B–D**) show that all inhibitor complexes share a bound water (W1) bridging the central scaffold to the gatekeeper residue in ERK2 (Q103). Left panels show that a bound water (W2) bridges active site residues K52 and D165 in complexes with VTX11e, inhibitor #8, and inhibitor #16, but not GDC0994. (**E**) Overlay of GDC0994 (gray), inhibitor #8 (slate) and inhibitor #16 (cyan). The relative N-lobe movements in panels (**C**) and (**D**) may be explained by the differential hydrogen bonding of K52 and W1 to the triazolopyridine central scaffold of inhibitors #8 and #16, distinct from the pyridone scaffold of GDC0994. The position of the hydrogen bond of the triazole nitrogen with K52 relative to the pyridone oxygen moves the K52-E69 salt bridge in an outward direction in inhibitors #8 and #10 relative to GDC0994. Structures were superpositioned by aligning Cα atoms within the C-terminal domain (residues 109–141, 205–245, 272–310).

K52-E69 salt bridge that appeared common to the three R-state inhibitors. Conceivably, this might reflect new bonding interactions that favor greater compactness of the active site.

Taken together, the crystal structures showed only small conformational variations among these closely related ATP-competitive inhibitors. Nevertheless, certain features were common to each of the R-state inhibitors, involving Gly loop closure and outward movements of helix αC and αL16. These appeared to be favored either by differential contacts between the inhibitor and Y34, or different hydrogen bond configurations between the central scaffold and K52. In each case, coupling to helix αC was enabled by Y34-Y62 and K52-E69 residue interactions. In contrast, the bulky right-side substituent on GDC0994 that prevented Gly loop closure and movements of helix αC/αL16 correlated with retention of conformational exchange.

The results so far revealed correspondence in R-state selection between the amido-linked pyrrole scaffold of VTX11e and BVD523 and the triazolopyridine scaffold of the new inhibitors. We asked what might happen, if the central scaffold were chosen to more closely resemble VTX11e, while choosing a right-side substituent similar to GDC0994. Tizakertib/ATG017/AZD0364 is an ATP-competitive ERK inhibitor with a tetrahydropyrrolodiazepenone core that resembles the amidopyrrole group in VTX11e and BVD523, and a right-side 3,4-fluorobenzyl group that resembles the 3-fluoro 4-chlorobenzyl group in GDC0994 (*Figure 10A*; *Flemington et al., 2021*; *Ward et al., 2019*). Interestingly, the X-ray structure of ATG017 complexed with 0P-ERK2 (*Figure 10B*, green; PDBID:6SLG, *Ward et al., 2019*) showed an open Gly loop, relative to complexes of BVD523 and VTX11e with 0P-ERK2 (shown for 0P-ERK2:BVD523, *Figure 10B*, light blue; PDBID:6GDQ, *Heightman et al., 2018*). This was due to the rotation of the 3,4-fluorobenzyl ring in ATG017 away from Y34 to form closer interactions with backbone atoms in the β1 and β2 strands. Unlike inhibitors #8 and #16, bound waters between the 3,4-fluorobenzyl ring and Y34 were absent, due to the open Gly loop conformation. As a result, Y34 shifts inwards along with helices αC and αL16, and the side chain of Y64 appears disordered (*Figure 10B*). The inward direction of this conformational shift suggested that ATG017 might share exchange properties with GDC0994 and the other pyridone-based inhibitors, despite their differences in central scaffold chemistry.

To examine this possibility, ATG017 was complexed with 2P-ERK2 and examined by NMR and HDX-MS. The NMR HMQC spectra revealed a small population shift to the R state, but with overall retention of R↔L exchange properties, similar to apoenzyme (*Figure 10C*). Likewise, HDX-MS showed little protection around the *P*+1 and helix αF segments (*Figure 10D*), with a pattern that was clearly distinct from the larger HDX protection induced by VTX11e or BVD523 (*Figure 3E and F*). Thus, ATG017 showed properties of conformational exchange and inward shifts of helices αC and αL16 that were shared with GDC0994 and inhibitors #4 and #15. This could be attributed to its right-side difluorobenzyl substituent that held the Gly loop in an open state. Interestingly, a bound water (W2) bridging K52 and D165 was present in the ATG017 complex (*Figure 10B*), and stronger HDX protection around the DFG segment and β9 strand was observed (*Figure 10D*; *Figure 10—figure supplement 1*). This implies that the HDX protection pattern around the DFG motif is functionally linked to

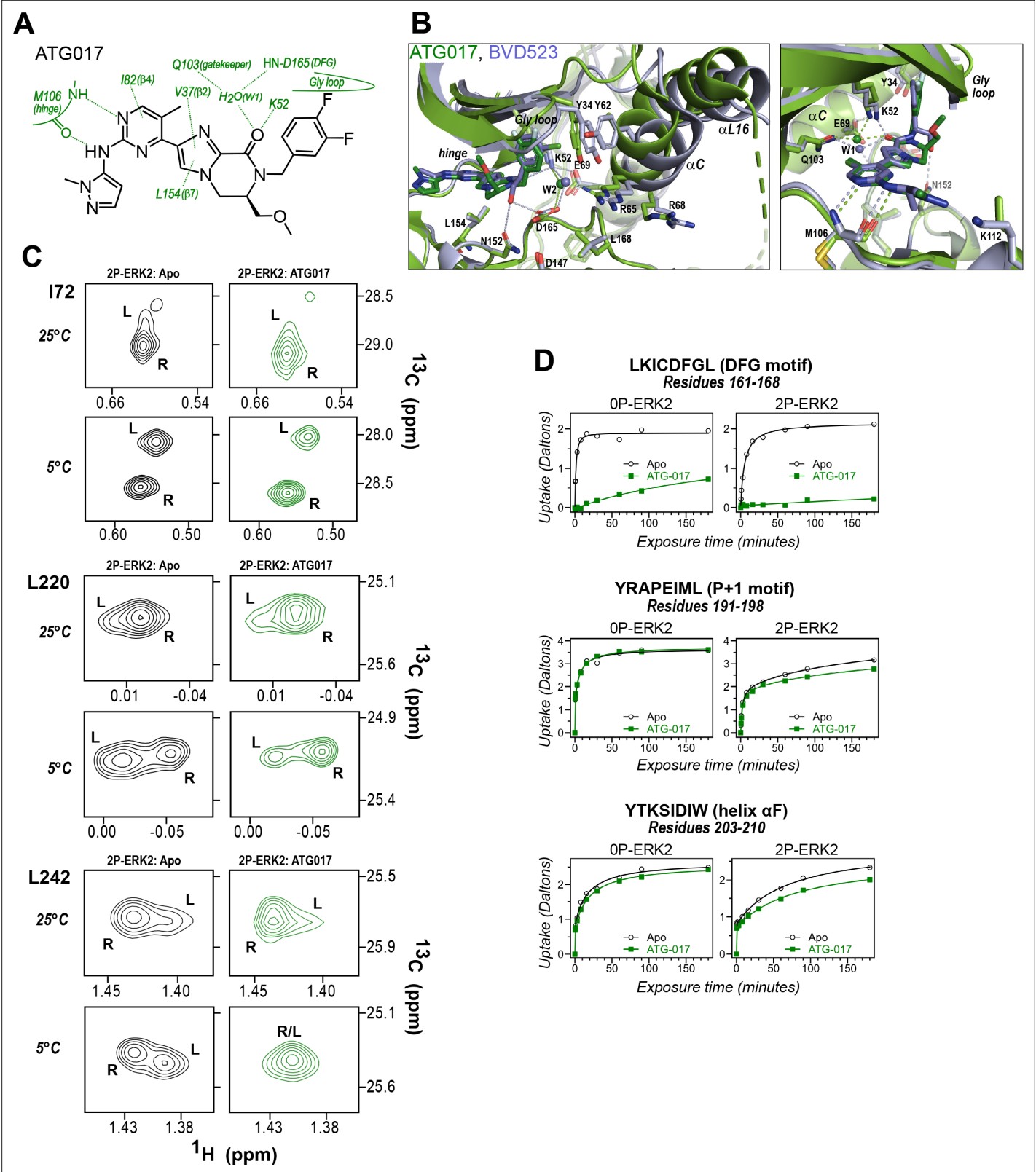

**Figure 10.** ATG017 promotes Gly loop opening and inward movement of N-lobe elements. (**A**) Summary of contacts formed by inhibitor ATG017 with active site residues in a co-crystal structure with 0P-ERK2 (PDBID:6SLG). (**B**) The active site of 0P-ERK2 complexed with ATG017 (PDBID:6SLG, green) and BVD523 (PDBID:6GDQ, light blue). *Left panel:* Front view showing movement of the Gly loop, helix αC, and helix αL16 in an outward direction by BVD523, attributed to the close proximity between the right-side chlorobenzyl substituent and the π orbital of Y34 in the Gly loop (Cl-π distance, 3.3 Å).

*Figure 10 continued on next page*

*Figure 10 continued*

*Right panel:* View showing left-side hydrogen bonds with main chain atoms of hinge residue M106. (**C**) 2D-HMQC spectra of 2P-ERK2 complexed with ATG017 at 25°C and 5°C, showing [methyl $^{13}$C,$^{1}$H] peaks of residues I72, L220 and L242. Unlike 2P-ERK2 complexed with BVD523, the ATG017 complex retains R↔L exchange resembling that of GDC0994 (*Figure 1*). (**D**) Hydrogen-deuterium exchange (HDX) time courses with ATG017 measuring deuterium uptake at the DFG motif, *P*+1 segment, and helix αF. Time courses for strand β9 are shown in *Figure 10—figure supplement 1*. Enhanced HDX protection by ATG017 binding is observed at the DFG and adjacent β9 segments, but minimally at the *P*+1 and helix αF, similar to that seen with GDC0994. The results suggest that allosteric coupling between the ligand binding pocket and distal regions surrounding the activation loop, but not the DFG motif or β9, are characteristic of R-state inhibitors. Structures were superpositioned by aligning Cα atoms within the C-terminal domain (residues 109–141, 205–245, 272–310).

The online version of this article includes the following figure supplement(s) for figure 10:

**Figure supplement 1.** Hydrogen-deuterium exchange (HDX) time courses responsive to ATG017 binding.

the water bridge (W2) linking K52 and D165. The results show that while changes around the DFG region may contribute to R-state selection, the movements of the Gly loop and helices αC and αL16 appear more essential for conformation selection and allosteric communication.

## Discussion

ERK inhibitors are promising therapeutic targets for cancers harboring oncogenic BRAF or RAS with acquired resistance to RAF and MEK inhibitors (*Ryan et al., 2015*; *Morris et al., 2013*; *Hatzivassiliou et al., 2012*; *Roskoski, 2019*; *Pan et al., 2022*). A feature of ERK2 activation that is somewhat unique among protein kinases is its global exchange behavior, involving millisecond interconversion between at least two states in its active, dual phosphorylated form. In this study, we present new ATP-competitive inhibitors of ERK1/2 that, despite their chemical divergence from VTX11e and BVD523, nevertheless share properties of R-state selection. They display sensitivity to the conformational differences between the R and L-states and the ability to induce long-distance perturbations at the *P*+1 segment and helix αL16. Crystallographic structures show correlations between the property of R-state selection and small but systematic shifts in the N-lobe that move helices αC and αL16 in the direction of the activation loop. This appears to be facilitated by movements of strands β1-β2-β3 coupled to helix αC. The findings reveal the importance of contacts between strand β3 and helix αC *via* the positioning of the K52-E69 salt bridge in ERK2, as well as the unique π−π contacts between Tyr residues in the Gly loop and helix αC, in guiding conformation selection for the R-state. By contrast, the binding of SCH772984 disrupts β3-αC contacts in ERK2 (*Chaikuad et al., 2014*), which may explain why this inhibitor favors the L-state (*Pegram et al., 2019*).

Unlike many other kinases, activation of ERK leads to relatively small structural changes within the active site. In 0P-ERK2, the N and C lobes are held apart by the backbone atoms of G167 in the DFG motif, which interdigitates with R65 and Q64 in helix αC (*Zhang et al., 1994*). This domain separation has been proposed to account for the low activity of unphosphorylated ERK2. Following phosphorylation, residue R65 rotates along with helix αC in the direction of the activation loop and pT183, partially relieving the barrier formed by G167. The resulting domain rotation between N- and C-lobes is only 5 degrees, but it brings K52 closer to D165 by 1.5 Å. The small magnitude of these changes are in part due to structural constraints unique to ERK2. Here, residues in the C-terminal L16 segment extending from the C-lobe to the N-lobe (e.g. Y316, D319, E320, F327, and M331) form close contacts with residues in helices αC and αE and the activation loop (*Figure 11A*). They culminate in helix αL16, where residues I345 and F346 fill a hydrophobic pocket formed by residues in helix αC and the β4-β5 loop (*Figure 11B*). This placement of helix αL16 aligns structurally with hydrophobic motif (HM) sequences at the C-termini of PKA and members of the AGC kinase subfamily (*Taylor et al., 2022*; *Kannan et al., 2007*; *Baffi and Newton, 2022*), as well as cyclins complexed with CDK (*Tatum and Endicott, 2020*). Thus helix αL16 coincides with regions in other kinases also involved in long-distance allostery.

Many of the key [methyl-$^{13}$C,$^{1}$H]-ILV probes that report global exchange behavior in ERK2 are part of the extensive network of hydrophobic residues connecting these substructures. Thus, the side chain of I345 fills a deep hydrophobic pocket formed by T66, L67, and I70 in helix αC, F57 in the β3-αC loop, V99 in strand β5, and M96 in loop β4-β5 (*Figure 11B*). This network also includes contacts formed by αL16 residue F346 with R89, I93, and M96 in the β4-β5 loop, and F352 with L74, I84, and I87 in αC and β4. Furthermore, side chains from F327 and M331 in L16 fill a hydrophobic pocket formed by

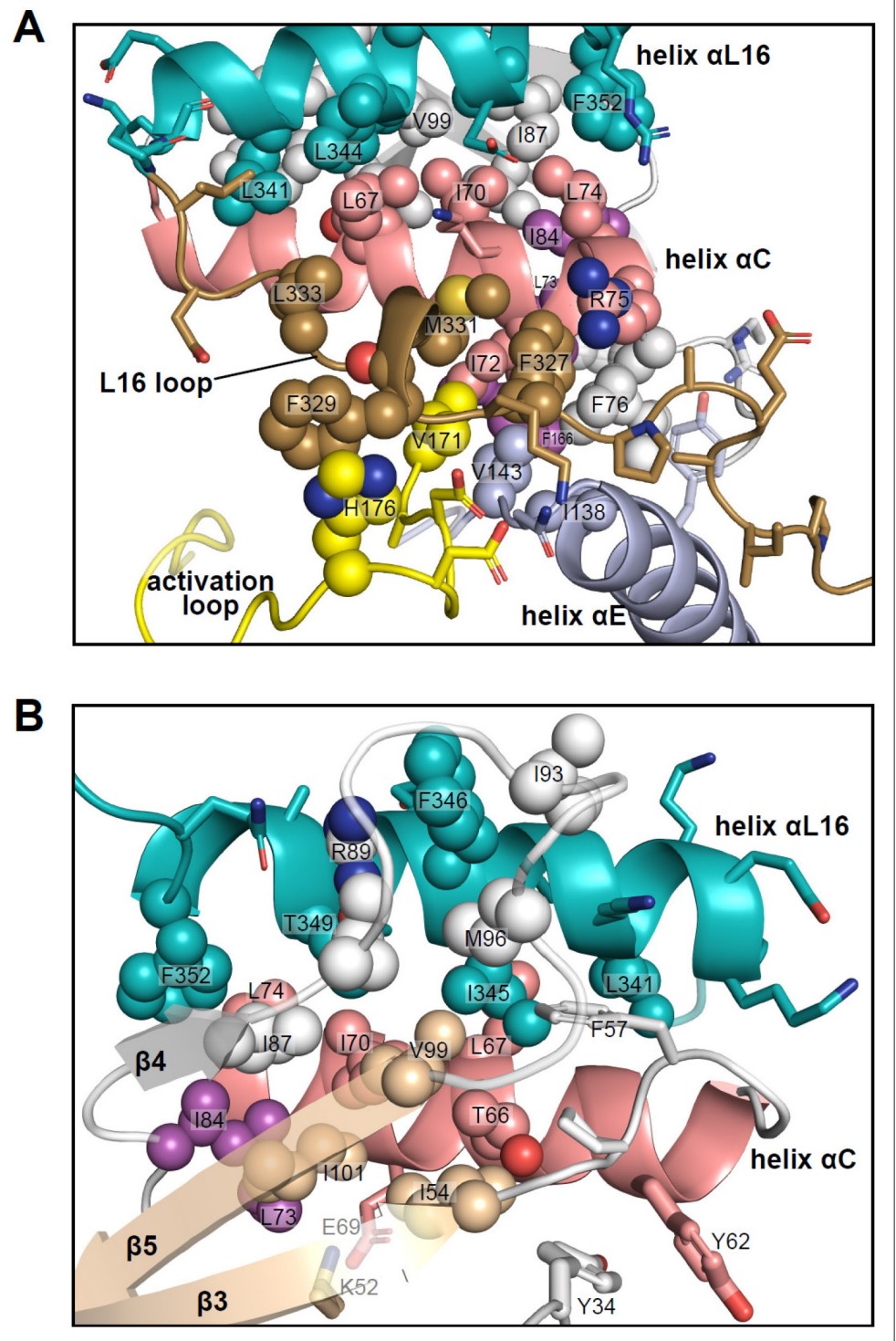

**Figure 11.** Interactions of L16 and helix αL16 with N-lobe elements. Views of 2P-ERK2 (PDBID:2ERK), showing hydrophobic residue interactions between (**A**) loop L16 (M331, F327) and helices αC and αE (I72, R75, F76, I138). Residues in helix αC in turn form second sphere interactions with residues nearby catalytic residues in β3 (I54, nearby K52), β8-β9 (F166, DFG motif), and β6 (V143, nearby HRD). (**B**) Residue interactions showing close connections between helix αL16 (I345, F346, T349, F352), and helix αC (I54, F57, T66, L67, I70, L74, I84, I87) and loop β4-β5 (R89, I93, M96, V99). Residues in magenta (L73, I84, F166) participate in the R-spine in ERK2.

The online version of this article includes the following figure supplement(s) for figure 11:

**Figure supplement 1.** Crystal contacts with the activation loop in X-ray structures of extracellular signal-regulated kinase-2 (ERK2).

I72, R75, and F76 in helix αC, V143 in the αE-β6 loop, and V171 in the activation loop (*Figure 11A*). In turn, I72 and F76 form hydrophobic contacts with L73 in helix αC, I138 in helix αE, and F166 in the DFG motif. Thus, αL16 and L16 contact the N-lobe *via* a hydrophobic network that merges with the R-spine (L73, I84, F166). Such extensive packing interactions may help to constrain ERK2, preventing the backbone rotational movements needed to allow a DFG-out configuration (*Vijayan et al., 2015*), which is never observed in the wild-type kinase. Conceivably, the conformations involved in R↔L global exchange could reflect systematic movements within this network, which propagate to produce the chemical shift perturbations observed in I345. Although such conformational shifts are small, the coupling between movements of the Gly loop and helices αC and αL16 are complementary to the N-lobe movements seen upon ERK2 phosphorylation. It is remarkable that inhibitors that share similar binding interactions and active site contacts are sensitive to such small conformational differences.

This study shows how HDX-MS and NMR are sensitive to slight perturbations within the active site that lead to significant long-distance responses at the activation loop and *P*+1 segment. Large structural rearrangements around the activation loop were not obvious in the X-ray structures of 2P-ERK2 complexed with GDC0994, VTX11e, or inhibitors #8 and #16. However, evidence that these measurements truly report conformational changes around pT183 and pY185 was supported by the ability of VTX11e to inhibit their rate of dephosphorylation by MAP kinase phosphatase (MKP3/DUSP6, *Pegram et al., 2019*). This demonstrates the protection of the activation loop by VTX11e binding. Conceivably, the conformation of the activation loop in solution may be more variable than in crystal structures of 2P-ERK2, in part due to lattice restraints. This is suggested by the extensive lattice contacts with the activation loop and *P*+1 segment in complexes formed by VTX11e, GDC0994, and inhibitors #8 and #16 (*Figure 11—figure supplement 1*). In fact, crystal contacts are positioned in a manner that might influence the activation loop in most X-ray structures of ERK2 (*Pegram et al., 2023*). This highlights the need for solution measurements to more fully understand allosteric regulation in ERK2.

Recently, we conducted extended molecular dynamics simulations (225 μs) starting from the X-ray structure of 2P-ERK2 (PDBID: 2ERK) (*Pegram et al., 2023*). The results showed unexpected flexibility of the activation loop, which deviated from the X-ray fold in all trajectories and was able to adopt multiple long-lived conformational states with RMSF ≤1.2 Å and lasting for more than 5 μs. Each state appeared only after 1–5 μs, perhaps due to biases introduced by lattice contacts in the starting model. Importantly, the states varied significantly in the contacts formed between the activation loop, L16, and the *P*+1 segment. These variable activation loop conformations were associated with differential dynamics within the active site, supporting allosteric coupling between these regions. Interestingly, in one of the novel states, the Arg salt bridges formed between pY185 and the *P*+1 segment were broken, allowing the phosphorylated side chain to move away and form a new salt bridge with helix αC. This reorganization resembled that observed in a co-crystal structure of 2P-ERK2 and the adaptor protein, PEA15 (*Mace et al., 2013*). The results suggest that distinct conformational states of the activation loop appear thermally accessible and likely to exist in solution. We speculate that they may function to accommodate variable binding modes for substrates and effectors. To the extent this is true, the coupling between the active site and activation loop in ERK2 might allow conformation selective inhibitors to control the binding of substrates and regulatory effectors. Similarly, allosteric properties in PKA allow different ATP-competitive inhibitors to vary the cooperativity of substrate binding (*Olivieri et al., 2022*).

Indeed, remote conformational changes induced by conformation selective inhibitors have been shown to alter noncatalytic scaffolding functions of protein kinases *via* interactions with regulatory enzymes (*Agudo-Ibáñez et al., 2023*; *Leroux and Biondi, 2020*; *Fang et al., 2020*; *Sonti et al., 2018*; *Joseph et al., 2022*). In fact, an intriguing recent study demonstrates that ubiquitination and degradation of 2P-ERK2 can be enhanced by BVD523, suggesting that conformation selective inhibitors may act as 'kinase degraders' that preferentially target the activated kinase (*Balmanno et al., 2023*). Other noncatalytic functions of ERK2 include the activation of topoisomerase IIα and poly-ADP ribose polymerase *via* direct binding of the phosphorylated kinase (*Shapiro et al., 1999*; *Cohen-Armon et al., 2007*). Preferential binding of DNA to 2P-ERK2 has also been shown to function in ERK1/2 binding to chromatin DNA, with a proposed regulation of transcription through phosphorylation-independent mechanisms (*Hu et al., 2009*; *Lawrence et al., 2008*; *McReynolds et al., 2016*; *Tee et al., 2014*). Thus, the ability of R-state inhibitors to control solution conformations of the activation loop may influence emerging noncatalytic functions of ERK.

In summary, the case of ERK2 illustrates how very small conformational changes in protein kinases can lead to considerable effects on activity and inhibition. Correspondingly, small variations between inhibitors which otherwise share similar binding interactions and active site contacts, differentially control long-distance allosteric effects that induce conformational changes at the activation loop. Such differences may be important for cellular responses, by altering the conformation of ERK at a key interface for substate and effector recognition. This highlights the importance of understanding the properties of conformation selection in ERK, and how these unique features of ERK regulation impact inhibitor design. The elements of protein architecture that enable small active site movements to yield large effects in distal regions remain an important and exciting feature of MAP kinases to explore in future investigations.

## Materials and methods

### Reagents and purified proteins

VTX11e, GDC0994, and ATG017 were obtained from Selleck Chemicals (Cat# S7709, S7554, S8708; Houston, TX). Dimethylsulfoxide (DMSO), dithiothreitol (DTT), formic acid, and $D_2O$ (99.9%, Cat#756882) were obtained from Sigma-Aldrich (Burlington, MA). Ni-NTA agarose was obtained from Qiagen (Cat#30210, Germantown, MD). Inhibitors #1-#17 were synthesized in-house at Genentech, Inc, following methods described in WIPO Patents WO2015103137 A1 20150709 and WO2013130976 A1 20130906.

### NMR spectroscopy

[methyl-$^{13}C$,$^{1}H$]-Ile, Leu, and Val-labeled rat His$_6$-ERK2 was prepared as described (*Xiao et al., 2014*; *Xiao et al., 2015*) in buffer containing 50 mM Tris-d11 pH 7.4 ($D_2O$), 150 mM NaCl, 5 mM MgSO$_4$, 0.1 mM EDTA, 1 mM D,L-1,4-DTT-d10, 100%(v/v) $D_2O$, and 2.5% (v/v) glycerol-d8. Stock solutions (25 mM) of all inhibitors (VTX11e, BVD523, GDC0994, ATG017, Inhibitors #1, #4, #5, #6, #8, #15, #16) were prepared in DMSO-d6, and added to ERK2 (150 μM) to form complexes with 100% binding stoichiometry ([ERK2]:[inhibitor]=1.0:1.2). The DMSO-d6 concentration in the final protein sample was ~1% (v/v).

Two-dimensional (2D) ($^{13}C$,$^{1}H$) heteronuclear multiple quantum coherence (HMQC) spectra of 0P- and 2P-ERK2 were collected on a Varian 900 MHz NMR spectrometer. Data were collected at 25°C and 5°C each for a total time of 10 hr. Each spectrum was acquired with nonuniform sampling (NUS) with 160 complex points (900 MHz) in the $t_1$ ($^{13}C$) dimension, corresponding to 29.1 ms at 900 MHz, and 1024 complex points in the acquisition period. WURST40 $^{13}C$ decoupling was applied during the 85 ms acquisition period, and a 1.5 s delay period was used between each scan. The spectral processing was performed with the software package NMRPipe (*Delaglio et al., 1995*). Time domain data in the $^{1}H$ dimension were apodized by a cosine window function and zero-filled prior to Fourier transformation. The indirect dimension ($^{13}C$) was apodized by a cosine window function and zero-filled prior to Fourier transformation. NUS spectral reconstruction utilized the SMILE plugin (*Ying et al., 2017*). Spectral visualization and analysis were achieved using CCPNMR Analysis software (*Vranken et al., 2005*). NMR spectra of 0P-ERK2 and 2P-ERK2 apoenzymes were collected in two or more experiments on separate enzyme preparations with nearly identical results. Spectra of ERK2 complexed with inhibitors were each collected in one experiment.

### X-ray crystallography

X-ray structure images were created with Pymol 2.3.4 software (Schrödinger LLC, New York, NY). Co-crystals of human His$_6$-2P-ERK2 in complex with Inhibitors #8 and #16 were grown at 20 °C in vapor diffusion hanging drops, mixing 0.5 μL protein +0.5 μL well solution (32–36% (w/v) PEG3350, 0.1 M Tris pH 8.8–9.0). Crystals were cryoprotected in well solution +10%(v/v) glycerol prior to data collection. X-ray diffraction data were collected on a Rigaku FR-E generator (Rigaku Americas Corporation, The Woodlands, TX) equipped with a Pilatus 2 M detector (Dectris USA, Philadelphia, PA). Data were processed using the Mosflm and Aimless programs in the CCP4 suite (*Winn et al., 2011*). Structures of ERK2 complexed with inhibitors were each collected in one experiment.

### Kinase inhibition assays

Kinetic parameters ($K_i$) were determined from aggregate kinase assays of human 2P-ERK2 performed with the fluorescent Omnia peptide substrate, S/T17 (Cat. KNZ1171C, Invitrogen, Carlsbad, CA).

Omnia assays contained 50 mM HEPES, pH 7.3, 10 mM MgCl$_2$, 1 mM dithiothreitol, 0.005% Triton-X100, 5 nM ERK2, 6.25µM S/T17 peptide substrate, and 25 µM ATP (corresponding to the observed K$_m$) in a total reaction volume of 25 µL. Assays were run at ambient temperature in 384-well plates, collecting time points every 50 s for 30 min on an Envision plate reader (PerkinElmer, Waltham, MA) with excitation 340 nm and emission 495 nm. Phosphorylation rates were normalized to controls (no inhibitor) and plotted against inhibitor concentration to obtain IC$_{50}$ values using a four-parameter fit.

## HDX-MS

Wild-type rat 0P-ERK2 and 2P-ERK2 were prepared as described, after bacterial expression from a plasmid constructed by Marcelo Sousa and Sandra Metzner, U. Colorado, Boulder (*Pegram et al., 2019*). ERK2 protein (17 µM) was incubated with 20 µM inhibitor (VTX11e, BVD523, GDC0994, ATG017, Inhibitors #1-#17) for 30 min in HDX buffer (50 mM potassium phosphate pH 7.2, 100 mM NaCl, 5 mM DTT, 10 mM MgCl$_2$ and 0.5% (v/v) DMSO from concentrated ligand solutions). Deuterium uptake was initiated by adding 54 µL D$_2$O (containing an equal concentration of ligand, DMSO, and MgCl$_2$) to 6 µL ERK2 (complexed with or without ligand), to reach a final buffer concentration of 5 mM potassium phosphate pH 7.2, 10 mM NaCl, 0.5 mM DTT, 10 mM MgCl$_2$, and 0.5% DMSO. Reactions were incubated at 25.0 °C for varying times between 30 s and 180 min (0.5, 1, 3, 8, 16, 30, 60, 90, 180 min). Exchange was quenched with 48 µL 100 mM potassium phosphate pH 2.2. Additionally, an in-exchange control was performed by the addition of quenching solution prior to initiating deuterium uptake, with no incubation period. All reactions were immediately injected in 100 µL volumes onto a Waters nanoAcquity HDX Manager UPLC system for proteolysis on an immobilized pepsin column (Enzymate, Waters Corp, Milford, MA) at 10 °C. Proteolysis, as well as peptide desalting (Vanguard Acquity UPLC BEH C18), were carried out in isocratic Solvent A (0.1%(v/v) formic acid in H$_2$O) with flowrate 100 µL/min. Peptide separations (Waters Acquity UPLC BEH C18, 1.7 µm, 1.0 × 100 mm) were carried out at 3 °C using a 12 min linear gradient from 8–85% Solvent B (0.1% formic acid in acetonitrile) at 40 µL/min.

Peptides were analyzed by ESI-MS/MS using a Waters Synapt G2 HDMS Q-TOF mass spectrometer in positive ion mode. Continuum data were collected over 50–2000 m/z with 0.23 s scan time. Lock mass correction was achieved using Glu-Fibrinogen peptide (MH$_2^{+2}$ = 785.8426 m/z), with resolution = 20,000 at m/z 956. Undeuterated pepsin-cleaved peptides from 0P- or 2P-ERK2 were identified by MS$^e$ sequencing using PLGS 3.0 (Waters). Searches of rat ERK2 allowed nonspecific digestion and variable modifications of oxidized Met and phosphorylated Ser/Thr/Tyr, with parameter settings of intensity threshold = 750 counts, lock mass window = 0.4 Da, and low energy threshold = 135 counts. Peptides were accepted after filtering for high confidence, which required a maximum MH$^+$ error of 15 ppm, minimum sequence length 3, maximum sequence length 25, minimum product ions 3, and identification in at least 2 of 4 replicate runs. Deuterium uptake was calculated for high-confidence peptides in each hydrogen exchange dataset using DynamX 3.0 (Waters). Raw files were processed using a 0.35 Da lock mass window and a low energy threshold of 100. For every experiment, automated ion assignments for deuterated peptides were manually inspected and validated. The weighted average mass of isotope distributions was calculated for each verified peptide using DynamX, and referenced to the weighted average mass of its unlabeled form. The recorded uptake from the in-exchange control was subtracted from the uptake values of all timepoints (0.5, 1, 3, 8, 16, 30, 60, 90, 180 min) to yield corrected uptake measurements that were used in plots of time courses and dAUC calculations. *Supplementary file 1* contains deuterium uptake values without this correction. HDX time courses of 0P-ERK2 and 2P-ERK2 apoenzymes were collected in two or more experiments on separate enzyme preparations with nearly identical results. Time courses of ERK2 complexed with inhibitors were each collected in one experiment.

## Dataset availability

Crystallography data files have been deposited in the RSCB PDB under accession codes 8U8J and 8U8K. NMR datafiles have been deposited in BMRbig under accession code BMRbig96. Data files generated from HDX experiments have been deposited to the ProteomeXchange Consortium *via* the PRIDE partner repository (*Perez-Riverol et al., 2022*), with the dataset identifier PXD048311.

## Materials availability

Proprietary ERK inhibitors are available upon request from Dr. John Moffat, Genentech, Inc Recipients will be asked to sign a materials transfer agreement and will be restricted from distributing these compounds or use them in any applications related to for-profit research. Plasmids for the expression of ERK2 and constitutively active MKK1-G7B (*Xiao et al., 2014*; *Xiao et al., 2015*; *Pegram et al., 2019*) are available upon request from Dr. Natalie Ahn, University of Colorado - Boulder.

## Acknowledgements

We are indebted to Drs. Marcelo Sousa, Elan Eisenmesser, and Johannes Rudolph for guidance on structural analyses and many valuable discussions, and Josh Anderson for insightful discussions about data analysis and presentation. We also thank Drs. Thomas Lee, Chris Ebmeier, and Rachel Mehaffey for mass spectrometric analyses of purified proteins, Chang Liu for preliminary HDX studies, and Daniel Lee for Python script coding for HDX datasets. Special thanks to the Array Structural Biology group for their valuable contributions to X-ray structure determination, and to Dr. Yao Xiao for NMR inhibitor binding titrations in this study. This work was supported by NIH Research Awards R35GM136392 and R01GM114594 (NGA), NIH Instrumentation Awards S10RR026641 and S10OD025267 (mass spectrometry) and S10OD025020 (NMR), and Cancer Center Support Grant P30CA046934 (CU Anschutz NMR Shared Resource Facility). Work on the development and synthesis of ERK inhibitors and determination of X-ray structures was funded by Genentech, Inc.

## Additional information

### Competing interests

Guy P Vigers: Guy Vigers is Structural Biology Consultant at Allium Consulting LLC. Huifen Chen: Dr. Huifen Chen is Distinguished Scientist at Genentech, Inc. John G Moffat: Dr. John Moffat is Distinguished Scientist at Genentech, Inc. The other authors declare that no competing interests exist.

### Funding

| Funder | Grant reference number | Author |
| --- | --- | --- |
| National Institutes of Health | R35GM136392 | Natalie G Ahn |
| National Institutes of Health | R01GM114594 | Natalie G Ahn |
| National Institutes of Health | S10RR026641 | Natalie G Ahn |
| National Institutes of Health | S10OD025267 | Natalie G Ahn |
| National Institutes of Health | S10OD025020 | David N Jones |
| National Institutes of Health | P30CA046934 | David N Jones |

The funders had no role in study design, data collection and interpretation, or the decision to submit the work for publication.

### Author contributions

Jake W Anderson, Conceptualization, Data curation, Formal analysis, Validation, Investigation, Visualization, Methodology, Writing – original draft, Writing – review and editing; David Vaisar, Data curation, Formal analysis, Investigation, Methodology, Writing – review and editing; David N Jones, Huifen Chen, John G Moffat, Conceptualization, Resources, Data curation, Formal analysis, Validation, Investigation, Methodology, Writing – review and editing; Laurel M Pegram, Conceptualization, Validation, Investigation, Methodology, Writing – review and editing; Guy P Vigers, Data curation, Formal analysis, Validation, Methodology, Writing – review and editing; Natalie G Ahn, Conceptualization,

Resources, Data curation, Formal analysis, Supervision, Funding acquisition, Validation, Investigation, Visualization, Methodology, Writing – original draft, Project administration

### Author ORCIDs
Jake W Anderson ⓘD https://orcid.org/0000-0001-8757-7408
Natalie G Ahn ⓘD https://orcid.org/0000-0002-2690-2630

Reviewer #1 (Public Review): https://doi.org/10.7554/eLife.91507.3.sa1
Reviewer #2 (Public Review): https://doi.org/10.7554/eLife.91507.3.sa2
Reviewer #3 (Public Review): https://doi.org/10.7554/eLife.91507.3.sa3
Author Response https://doi.org/10.7554/eLife.91507.3.sa4

## Additional files

### Supplementary files
- Supplementary file 1. Deuterium uptake time courses for all HDX-MS experiments.
- Supplementary file 2. NMR chemical shifts and perturbations by selected inhibitors.
- Supplementary file 3. X-ray data collection and refinement parameters.
- MDAR checklist

### Data availability
Crystallography data have been deposited in the PDB under accession codes 8U8J and 8U8K. Data generated from HDX experiments, including Supplementary Dataset S1, coverage maps, and representative raw spectra have been deposited to the ProteomeXchange Consortium via the PRIDE partner repository *Perez-Riverol et al., 2022*, with the dataset identifier PXD048311. Data generated from NMR experiments have been deposited in BMRbig under accession code BMRbig96.

The following datasets were generated:

| Author(s) | Year | Dataset title | Dataset URL | Database and Identifier |
|---|---|---|---|---|
| Anderson JW, Vaisar D | 2024 | HDX-MS Analysis of Conformation Selection by ATP-competitive Inhibitors and Allosteric Communication in ERK2 | https://www.ebi.ac.uk/pride/archive/projects/PXD048311 | PRIDE, PXD048311 |
| Anderson JW, Jones DN | 2024 | NMR Analysis of Chemical Shift Perturbations by ATP-competitive Inhibitors complexed with ERK2 | https://bmrbig.org/released/bmrbig96 | BMRbig, BMRbig96 |
| Anderson JW, Vigers GP | 2024 | Co-crystal structure of phosphorylated ERK2 in complex with ERK1/2 inhibitor #16 | https://www.rcsb.org/structure/8U8J | RCSB Protein Data Bank, 8U8J |
| Anderson JW, Vigers GP | 2024 | Co-crystal structure of phosphorylated ERK2 in complex with ERK1/2 inhibitor #8 | https://www.rcsb.org/structure/8U8K | RCSB Protein Data Bank, 8U8K |

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
