## [Editor Report · eLife assessment]

This **fundamental** study provides **compelling** evidence to explain how chemical variations within a set of kinase inhibitors drive the selection of specific Erk2 conformations. Conformational selection plays a critical role in targeting medically relevant kinases such as Erk2 and the findings reported here open new avenues for designing small molecule inhibitors that block the active site while also steering the population of the enzyme into active or inactive conformations. Since protein dynamics and conformational ensembles are essential for enzyme function, this work will be of broad interest to those working in drug development, signal transduction, and enzymology.

---

## [Referee Report · Reviewer #1 (Public Review)]

Summary:

The authors set out to determine how chemical variation on kinase inhibitors determines selection of Erk2 conformations and how inhibitor binding affects ERk2 structure and dynamics.

Strengths:

The study is beautifully presented both verbally and visually. The NMR experiments and the HDX experiments complement each other for the study of Erk2 solution dynamics. X-ray crystallography of Erk2 complexes with inhibitors show small but distinct structural changes that support the proposed model for the impact of inhibitor binding.

---

## [Referee Report · Reviewer #2 (Public Review)]

Erk2 is an essential element of the MAP kinase signaling cascade and directly controls cell proliferation, migration, and survival. Therefore, it is one of the most important drug targets for cancer therapy. The catalytic subunit of Erk2 has a bilobal architecture, with the small lobe harboring the nucleotide-binding pocket and the large lobe harboring the substrate-binding cleft. Several studies by the Ahn group revealed that the catalytic domain hops between (at least) two conformational states: active (R) and inactive (L), which exchange in the millisecond time scale based on the chemical shift mapping. The R state is a signature of the double phosphorylated Erk2 (2P-Erk2), while the L state has been associated with the unphosphorylated kinase (0P-Erk2). Interestingly, the X-ray structures reveal only minimal differences between these two states, a feature that led to the conclusion that active and inactive states are structurally similar but dynamically very different. The Ahn group also found that ATP-competitive inhibitors can steer the populations of Erk2 either toward the R or the L state, depending on their chemical nature. The latter opens up the possibility of modulating the activity of this kinase by changing the chemistry of the ATP-competitive inhibitor. To prove this point, the authors present a set of nineteen compounds with diverse chemical substituents. From their combined NMR and HDX-Mass Spec analyses, fourteen inhibitors drive the kinase toward the R state, while four compounds keep the kinase hopping between the R and L states. Based on these data, the authors rationalize the effects of these inhibitors and the importance of the nature of the substituents on the central scaffold to steer the kinase activity. While all these inhibitors target the ATP binding pocket, they display diverse structural and dynamic effects on the kinase, selecting a specific structural state. Although the inhibited kinase is no longer able to phosphorylate substrates, it can initiate signaling events functioning as scaffolds for other proteins. Therefore, by changing the chemistry of the inhibitors it may be possible to affect the MAP cascade in a predictable manner. This concept, recently introduced as proof of principle, finds here its significance and practical implications. The design of the next-generation inhibitors must be taken into account for these design principles.

The research is well executed, and the data support the author's conclusions.

---

## [Referee Report · Reviewer #3 (Public Review)]

Summary:

Anderson at al utilize an array of orthogonal techniques to highlight the important of protein dynamics for the function and inhibition of the kinase ERK2. ERK2 is important for a large variety of biological functions.

Strengths:

This is a thorough and detailed study that uses a variety of techniques to identify critical molecular/chemical parameters that drive ERK2 in specific states.

Weaknesses:

No details rules were identified so that novel inhibitors could be designed. Nevertheless, the mode of action of these existing inhibitors are much better defined.

---

## [Author Response]

The following is the authors’ response to the original reviews.

**eLife assessment**
This fundamental study provides compelling evidence to explain how chemical variations within a set of kinase inhibitors drive the selection of specific Erk2 conformations. Conformational selection plays a critical role in targeting medically relevant kinases such as Erk2 and the findings reported here open new avenues for designing small molecule inhibitors that block the active site while also steering the population of the enzyme into active or inactive conformations. Since protein dynamics and conformational ensembles are essential for enzyme function, this work will be of broad interest to those working in drug development, signal transduction, and enzymology.
**Public Reviews:**

**Reviewer #1 (Public Review):**
Summary: The authors set out to determine how chemical variation on kinase inhibitors determines the selection of Erk2 conformations and how inhibitor binding affects ERk2 structure and dynamics.Strengths: The study is beautifully presented both verbally and visually. The NMR experiments and the HDX experiments complement each other for the study of Erk2 solution dynamics. X-ray crystallography of Erk2 complexes with inhibitors shows small but distinct structural changes that support the proposed model for the impact of inhibitor binding.Weaknesses: A discussion of compound residence time for the different compounds and kinase constructs and how it could affect the very slow HDX rates might be helpful. For example, could any of the observed effects in Figure 4 be due to slow compound dissociation rather than slowed down kinase dynamics? What would be the implications?

Response: Rate constants for kon and koff were estimated for three inhibitors using surface plasmon resonanceAuthor response table 1:

**Author response table 1. sa4table1:** SPR estimates of Kd for selected inhibitors ranged between 0. 03-3 nM. All HDX time courses involved prebinding of 20 µM inhibitor and 17 µM ERK2 for 30 min (predicted occupancy 99.9%), followed by deuteration time courses with 20 µM inhibitor and 1.7 µM ERK2. Estimated rates of dissociation were ~0.0003-0.007 s-1 and rates of binding were 20-100 s-1 for the inhibitors tested. Because the binding rates are faster than the intrinsic H-D exchange rate at pD 7 (~1 s-1), we expect ligands to rebind and form the enzyme:ligand complex faster than the free enzyme undergoes exchange. Therefore, HDX rates should mostly reflect deuteration of the inhibitor-bound enzyme for all inhibitors.

	2P-ERK2	0P-ERK2
k_on_ (M^–1^s ^–1^)	k_off_ (s ^–1^)	k_d_ (nM)	k_on_ (M^–1^s ^–1^)	k_off_ (s ^–1^)	k_d_ (nM)
Inh#13	4.9 E+06	7.4 E-03	1.5	0.83 E+06	2.5 E-03	3.0
Inh#15	1.4 E+06	0.30 E-03	0.21	0.70 E+06	0.62 E-03	0.88
Inh#6	3.2 E+06	0.67 E-03	0.31	1.60 E+06	3.0 E-03	1.9

**Reviewer #2 (Public Review):**
Erk2 is an essential element of the MAP kinase signaling cascade and directly controls cell proliferation, migration, and survival. Therefore, it is one of the most important drug targets for cancer therapy. The catalytic subunit of Erk2 has a bilobal architecture, with the small lobe harboring the nucleotide-binding pocket and the large lobe harboring the substrate-binding cleft. Several studies by the Ahn group revealed that the catalytic domain hops between (at least) two conformational states: active (R) and inactive (L), which exchange in the millisecond time scale based on the chemical shift mapping. The R state is a signature of the double phosphorylated Erk2 (2P-Erk2), while the L state has been associated with the unphosphorylated kinase (0P-Erk2). Interestingly, the X-ray structures reveal only minimal differences between these two states, a feature that led to the conclusion that active and inactive states are structurally similar but dynamically very different. The Ahn group also found that ATP-competitive inhibitors can steer the populations of Erk2 either toward the R or the L state, depending on their chemical nature. The latter opens up the possibility of modulating the activity of this kinase by changing the chemistry of the ATP-competitive inhibitor. To prove this point, the authors present a set of nineteen compounds with diverse chemical substituents. From their combined NMR and HDX-Mass Spec analyses, fourteen inhibitors drive the kinase toward the R state, while four compounds keep the kinase hopping between the R and L states. Based on these data, the authors rationalize the effects of these inhibitors and the importance of the nature of the substituents on the central scaffold to steer the kinase activity. While all these inhibitors target the ATP binding pocket, they display diverse structural and dynamic effects on the kinase, selecting a specific structural state. Although the inhibited kinase is no longer able to phosphorylate substrates, it can initiate signaling events functioning as scaffolds for other proteins. Therefore, by changing the chemistry of the inhibitors it may be possible to affect the MAP cascade in a predictable manner. This concept, recently introduced as proof of principle, finds here its significance and practical implications. The design of the next-generation inhibitors must be taken into account for these design principles. The research is well executed, and the data support the author's conclusions.
**Reviewer #3 (Public Review):**
Summary: Anderson et al utilize an array of orthogonal techniques to highlight the importance of protein dynamics for the function and inhibition of the kinase ERK2. ERK2 is important for a large variety of biological functions.Strengths: This is a thorough and detailed study that uses a variety of techniques to identify critical molecular/chemical parameters that drive ERK2 in specific states.Weaknesses: No details rules were identified so that novel inhibitors could be designed. Nevertheless, the mode of action of these existing inhibitors is much better defined.

Response: As recommended we added a sentence to the Discussion suggesting that inhibitors that perturb the β1-β2-β3 sheet in such a way that moves helix αC and αL16 away from the binding site might confer R-state selection. We view this as a preliminary model for predicting conformation selection in ERK2.

**Reviewer #1 (Recommendations For The Authors):**
Maybe the authors can comment on how the HDX timescale and the NMR timescale relate to each other and how such different timescales can report on the same event. In particular, the HDX timescale appears to be on the scale on minutes to tens hours (e.g. 2P state). How would inhibitor dissociation and rebinding affect the observed HDX signal? Is it worth considering compound residence time for the different compounds/kinase states?

Response: The HDX-MS and NMR experiments report different processes therefore their timescales do not necessarily match. For native state proteins at neutral pH, HDX-MS reports fluctuations that allow solvent exposure of backbone amide N-H, reflecting conformational mobility of the main chain. This is often modeled as a two-state interconversion between “closed” (HDX protected) and “open” (HDX accessible) states. Because the µs-ms timescale of main chain fluctuations is faster than the intrinsic rate of HDX (kexch, ~1 s-1), the observed HDX rate (kobs) can be approximated by the ratio of kopen/kclosed x kexch = Kop x kexch. Therefore, kobs can be considered a thermodynamic measurement that reflects Kop.

The [methyl 13C,1H] NMR CPMG experiment that we used to identify global exchange behavior in Xiao et al (PNAS, 2014) modeled the 2P-ERK2 apoenzyme by a two-state equilibrium (L↔R) between methyl-ILV conformers, yielding rate constants kL→R 240 s-1 and kR→L 60 s-1. Some methyls had large enough chemical shifts between L and R that they appeared as separate peaks in HMQC spectra that matched the L and R populations estimated by CPMG. In this study, the HMQC peaks shown in Figures 1, 6, and 9 are those that report shifts in L vs R populations and conformation selection for the R-state by VTX11e, BVD523 and triazolopyridine inhibitors.

Where HDX and NMR agree is in their ability to report changes in populations of L and R in 2P-ERK2. This was first shown when both HDX and NMR measurements reported perturbations at the activation loop induced by inhibitors with differential selection for the R- vs L-states (Pegram et al. PNAS, 2019). CPMG measurements then confirmed that methyl probes in the activation loop are included in the global exchange process (Iverson et al., Biochemistry, 2020). Therefore, the HDX and NMR experiments reflect shifts in the equilibrium between L and R conformers, rather than motions with specific timescales.

**Reviewer #2 (Recommendations For The Authors):**
I believe the paper is suitable for the special issue of Elife dedicated to protein kinases after the authors address minor concerns/comments.a) Introduction, page 3: "[..] But within the ATP binding site, the conserved residues ...are largely overlapping." Do the authors mean that the residues are overlapping in the X-ray structures? If so, what is the rmsd among the X-ray structures?

Response: The overlap between conserved residues K52, E69, D147, N152 and D165 in 2P- and 0P-ERK2 is presented in Fig. S1C, which shows an overlay between their apoenzyme crystal structures (PDBID: 2ERK, 5UMO). The RMSD of atoms in each residue are: K52 0.63 Å (9 atoms); E69 0.15 Å (9 atoms); D147 0.055 Å (8 atoms); D165 0.88 Å (8 atoms). As recommended, this information was added to the legend to Suppl. Fig. S1.

b) Introduction, page 5: "[...] For example binding of VTX11 partially inhibits...[..]" Please provide a citation.

Response: As recommended we added a citation at end of this sentence (Pegram et al. PNAS, 2019).

c) Introduction, page 5: "[...] N-lobe deformities..." What do the authors mean by deformities? Are there frustrated conformations?

Response: We used the term “deformities” to mean conformational differences, which may be but are not necessarily due to frustration. To avoid confusion, we removed the term “deformities” and replaced it with “conformational changes”.

d) Supplementary Information. The authors report the chemical shift perturbations for several inhibitors. Does the extent of the chemical shift perturbation reflect the strength of the binding for each inhibitor? In other words, do the largest chemical shift perturbations correspond to the highest binding affinity?

Response: The concentrations used in the NMR ligand binding experiments (150 µM ERK2, 180 µM inhibitor) allow 99.9+% complex formation over the 0.03 - 3 nM range of Ki for all inhibitors. Therefore, the chemical shifts report changes in electronic environment between bound and free enzyme. These can be ascribed to first or second sphere contacts with ligand or distal allosteric effects. But they are not likely to reflect differences in binding affinity.

New Suppl. Fig. S3 now adds HMQC titrations of VTX11e and GDC0994 into 2P-ERK2, which confirm binding saturation based on the disappearance of free enzyme peaks.

e) Do the authors have any evidence for the dynamic effects of the different inhibitors? Of course, a systematic analysis of the protein dynamics by NMR will require a significant amount of time and effort beyond this work. However, did the authors measure the effects of the inhibitors on the linewidths of the methyl groups distal from the binding site?Response: As recommended, we examined linewidths of selected peaks in the presence and absence of inhibitors. The results show no significant systematic differences between bound and free ERK2. Therefore dynamic effects of different inhibitors are not indicated by the available data.f) The authors identified the b3-aC loop as a critical element for the internal network of interactions. Can this structural element be targeted by small molecules as well?

Response: Yes, in fact the X-ray structures of 0P-ERK2 bound to the inhibitor, SCH772984, and 2P-ERK2 bound to the related compound, SCHCPD336, both show inhibitor occupying a pocket between between strand β3 and helix αC, leading to disruption of β3-αC contacts (Chaikaud et al., NSMB 2014; Pegram et al., PNAS 2019). To the extent that β3-αC contacts are important for conformation selection to the R-state, this may explain why SCH772984 favors the L-state. We revised the Discussion to add this point.

g) The authors should mention a recent paper suggesting that it is possible to control substrate-binding affinity by changing the nature of the ATP-binding inhibitors ((DOI: 10.1126/sciadv.abo0696).

Response. As recommended we added this point and citation to the Discussion.

**Reviewer #3 (Recommendations For The Authors):**
3.1. The manuscript is well written, but very long and sometimes repetitive. Some parts of the introduction are repeated in the result section and parts of the result section are repeated in the discussion. It will be easy to shorten the work to make it easier to read.

Response: As recommended we streamlined the Discussion to remove some of the repetitive elements, while trying to retain the main conclusions and rationale for readers who are not well versed in kinase structure.

3.2. Only specific residues are shown for the NMR spectra figures - while this is helpful to understand the concept, full spectra need to be shown to allow for direct comparison of the data quality (i.e. in supplemental material). If statements are made that measurements are done under full saturation - it should be shown that saturation is achieved in the measurements. All relaxation data should be made available - similar to CSPs.

Response: As recommended, new Suppl. Figs. S2 and S9 were added to show the full spectra of each inhibitor complex analyzed by NMR. New Suppl. Fig. S3 now adds titrations of 2P-ERK2 with VTX11e and GDC0994.The results confirm binding saturation based on the disappearance of free enzyme peaks.

3.3. No validation report was provided, nor a PDB number - so it is unclear if the crystal structures have been submitted - they need to be submitted in order to also access an mtz file, which is critical to understanding the quality of the structure (especially the ligand). This makes it difficult to assess the quality of the structures.

Response: Table S1 has been revised to show data collection and refinement parameters for PDBID: 8U8K (2PERK2:Inh#8, Fig. 8C) and 8U8J (2P-ERK2:Inh#16, Fig. 8D). RCSB validation reports are attached and PDB depositions have been approved and will be released upon VOR assignment.